EMBO
Molecular Medicine

# Anti-tumor effects of PIM/PI3K/mTOR triple kinase inhibitor IBL-302 in neuroblastoma

Sofie Mohlin[1],[*],[‡] (iD), Karin Hansson[2],[‡], Katarzyna Radke[2], Sonia Martinez[3] (iD), Carmen Blanco-Apiricio[3], Cristian Garcia-Ruiz[2],[†] (iD), Charlotte Welinder[4], Javanshir Esfandyari[2], Michael O'Neill[5], Joaquin Pastor[3], Kristoffer von Stedingk[1],[6] & Daniel Bexell[2],[**] (iD)

## Abstract

The PI3K pathway is a major driver of cancer progression. However, clinical resistance to PI3K inhibition is common. IBL-302 is a novel highly specific triple PIM, PI3K, and mTOR inhibitor. Screening IBL-302 in over 700 cell lines representing 47 tumor types identified neuroblastoma as a strong candidate for PIM/PI3K/mTOR inhibition. IBL-302 was more effective than single PI3K inhibition *in vitro,* and IBL-302 treatment of neuroblastoma patient-derived xenograft (PDX) cells induced apoptosis, differentiated tumor cells, and decreased N-Myc protein levels. IBL-302 further enhanced the effect of the common cytotoxic chemotherapies cisplatin, doxorubicin, and etoposide. Global genome, proteome, and phosphoproteome analyses identified crucial biological processes, including cell motility and apoptosis, targeted by IBL-302 treatment. While IBL-302 treatment alone reduced tumor growth *in vivo*, combination therapy with low-dose cisplatin inhibited neuroblastoma PDX growth. Complementing conventional chemotherapy treatment with PIM/PI3K/mTOR inhibition has the potential to improve clinical outcomes and reduce severe late effects in children with high-risk neuroblastoma.

**Keywords** cisplatin; IBL-302; multikinase inhibition; neuroblastoma; PI3K
**Subject Categories** Cancer; Neuroscience; Chemical Biology

## Introduction

Phosphoinositol-3-kinase (PI3K) activation is a driver in many cancers (Engelman, 2009). The mammalian target of rapamycin (mTOR), a downstream PI3K pathway effector, promotes the expression of proteins involved in cell growth and survival, and mTOR kinase inhibitors, either alone or in combination with PI3K inhibitors, have shown preclinical promise (Fruman *et al*, 2017). These encouraging data have led to clinical trials of small-molecule PI3K and PI3K/mTOR pathway inhibitors (Fruman *et al*, 2017).

Neuroblastoma is a childhood malignancy of the sympathetic nervous system that accounts for 15% of all pediatric cancer fatalities. Children with high-risk disease have poor survival rates, with up to 40% 5-year mortality. Current treatment strategies include high-dose chemotherapy, surgery, radiotherapy, and anti-GD2 therapy. We and others have shown that targeting the PI3K/mTOR pathway could be a viable treatment for aggressive neuroblastoma (Chesler *et al*, 2006; Johnsen *et al*, 2008; Segerström *et al*, 2011; Chanthery *et al*, 2012; Mohlin *et al*, 2013, 2015; Cage *et al*, 2015; Stewart *et al*, 2015; Vaughan *et al*, 2016). Several studies reported anti-tumor effects and improved survival rates in preclinical neuroblastoma models using PI3K inhibitors LY294002 (Chesler *et al*, 2006), PI-103 (Segerström *et al*, 2011), dactolisib/NVP-BEZ235 (Chanthery *et al*, 2012; Stewart *et al*, 2015; Vaughan *et al*, 2016), ZSTK474 (Vaughan *et al*, 2016), PIK-75 (Cage *et al*, 2015), PW-12 (Cage *et al*, 2015), BKM120 (Stewart *et al*, 2015), and/or SF1126 (Erdreich-Epstein *et al*, 2016). In addition, AKT inhibitor perifosine (Li *et al*, 2010, 2011) and mTOR inhibitors rapamycin (Johnsen *et al*, 2008), CCI-779 (Johnsen *et al*, 2008), and Torin (Vaughan *et al*, 2016) were demonstrated to decrease proliferation *in vitro* and consequent tumor growth *in vivo*. One explanation to the promising preclinical results from inhibiting PI3K/AKT/mTOR could be the anti-angiogenic downstream effects (Johnsen *et al*, 2008; Chanthery *et al*, 2012; Mohlin *et al*, 2015). The majority of reports also showed that neuroblastoma cells with high expression of *MYCN* were more sensitive to PI3K inhibition and/or that treatment resulted in downregulated N-Myc protein levels *in vitro* and *in vivo* (Cage *et al*, 2015;

1 Division of Pediatrics, Department of Clinical Sciences, Lund University, Lund, Sweden
2 Department of Laboratory Medicine, Translational Cancer Research, Lund University Cancer Center, Lund University, Lund, Sweden
3 Experimental Therapeutics Programme, Spanish National Cancer Research Centre (CNIO), Madrid, Spain
4 Division of Oncology and Pathology, Department of Clinical Sciences Lund, Lund University, Lund, Sweden
5 Inflection Biosciences Ltd, Blackrock, Ireland
6 Department of Oncogenomics, University Medical Center, University of Amsterdam, Amsterdam, The Netherlands
   *Corresponding author. Tel: +46 462226439; E-mail: sofie.mohlin@med.lu.se
   **Corresponding author. Tel: +46 462226423; E-mail: daniel.bexell@med.lu.se
   †Present address: Hematology and Hemotherapy Research Group, IIS La Fe, Valencia, Spain
   ‡These authors contributed equally to this work

Chanthery et al, 2012; Chesler et al, 2006, Erdreich-Epstein et al, 2016, Johnsen et al, 2008; Vaughan et al, 2016). Importantly, the PI3K inhibitor perifosine has recently been tested in Phase I/Ib clinical trials with promising therapeutic effects and negligible toxicity in 46 neuroblastoma patients (Kushner et al, 2017; Matsumoto et al, 2017).

However, intrinsic resistance and acquired resistance to PI3K inhibitors are possible major obstacles to effective treatment with these agents. Resistance mechanisms include activation of downstream mTOR complexes and activation of other networked signaling pathways (Elkabets et al, 2013). The serine/threonine proviral insertion site in murine leukemia virus (PIM) kinases is overexpressed in many cancers and is associated with MYC overexpression and metastasis. Increased PIM1-3 expression has been linked to PI3K inhibitor resistance (Nawijn et al, 2011; Le et al, 2016). As a result, we synthesized the IBL-300 series multikinase inhibitors to specifically target PIM and PI3K to improve efficacy.

Here, we show that PIM, PI3K, and mTOR inhibitor IBL-302 demonstrated robust target specificity. In 707 cell lines across 47 tumor types, neuroblastoma was most sensitive to IBL-302 treatment, and of 16 neuroblastoma cell lines, IBL-302 was generally more effective than PI3K inhibitors alone. We used neuroblastoma patient-derived xenografts (PDXs) and cell lines to investigate the benefit of combination therapies directed toward PIM, PI3K, and mTOR pathways. Nanomolar concentrations of IBL-302 induced apoptosis and tumor cell differentiation and reduced N-Myc protein levels. IBL-302 potentiated the effect of three clinically used chemotherapeutic agents in vitro and low-dose cisplatin in vivo. Finally, RNA sequencing and mass spectrometry analyses of IBL-302-treated PDX cells demonstrated that multitarget treatment induced apoptosis and cell death while diminishing cell growth and motility. Adding multikinase PIM/PI3K/mTOR inhibitors to current treatments to lower administered doses of highly toxic chemotherapy could decrease complications later in life and improve outcomes in these young patients.

# Results

### High PIM3 expression is associated with poor outcome

We first investigated the potential roles of PIM isoforms in neuroblastoma. PIM1 and PIM3 were expressed in neuroblastoma cell lines and PDX-derived cell cultures (Appendix Fig S1A and B). High levels of PIM1 and PIM3 were also significantly associated with adverse neuroblastoma patient outcomes [Appendix Fig S1C (PIM1) and D (PIM3)].

### Neuroblastoma is highly sensitive to triple PIM/PI3K/mTOR inhibition

We then developed single compound multikinase inhibitors directed toward PIM, PI3K, and mTOR (covered under patent WO2012/156756). The precise synthetic structures of IBL-301 and IBL-302 are outlined in Fig 1A and B, respectively. Due to superior pharmaceutical profile over IBL-301 in vivo, the closely related analog IBL-302 was selected for further development. We also included the previously published dual PIM/PI3K inhibitor IBL-202 in our studies

(Crassini et al, 2018). Target efficiency toward PI3Kα, mTOR, PIM1, PIM2, and PIM3 displayed as IC50 (nM) for all three inhibitors is shown in Fig 1C.

In an initial screen for tumor type sensitivity to triple PIM/PI3K/mTOR inhibition by IBL-302, 707 cell lines derived from 47 different tumor types were tested by the Genomics of Drug Sensitivity in Cancer (GDSC) screening program (Yang et al, 2013). Selecting for tumor types represented by at least five cell lines, $GI_{50}$ values for IBL-302 were compared for the remaining 35 tumor types. Neuroblastoma was the most sensitive cancer to IBL-302 treatment (Fig 2A). To establish whether neuroblastoma sensitivity to IBL-302 was due to triple target inhibition or whether the effects were conferred by PI3K pathway inhibition, $GI_{50}$ values were compared between IBL-302 and the PI3K/mTOR inhibitors dactolisib, PI-103, and omipalisib, the mTOR inhibitor AZD8055, and the ribosomal S6 kinase (RSK)/PIM inhibitor SL0101 using GDSC data. Across a panel of 31 tumor types, neuroblastoma did not show particular sensitivity to any of the five PI3K-specific inhibitors (Appendix Fig S1E–I). Furthermore, analyzing $GI_{50}$ values across a panel of 16 neuroblastoma cell lines showed that the majority were highly sensitive to IBL-302 compared with PI3K inhibitors alone (IBL-302 was among the top-two most sensitive compounds in 14 of the 16 cell lines; Fig 2B).

### Neuroblastoma cells differentiate in response to PIM/PI3K/mTOR inhibition

To assess the effects of IBL inhibitors, we treated neuroblastoma PDX cells (Appendix Table S1, and Braekeveldt et al, 2015; Persson et al, 2017) and conventional neuroblastoma cell lines with increasing drug concentrations surrounding their respective $GI_{50}$ range (Appendix Table S2). We first verified that IBL-202 (PIM/PI3K) and IBL-301 (PIM/PI3K/mTOR) targeted the intended individual signaling pathways. Treatment with the IBL inhibitors downregulated the levels of phosphorylated Akt (Ser473 and Thr308), phosphorylated p70S6K and p85S6K (Thr412 and Thr389, respectively), and phosphorylated PRAS40 (Thr246; Figs 3A–C and EV1A–C). It has been suggested that tumors are more susceptible to chemotherapy when induced into a differentiated state due to the reduction in the stem cell pool (Pietras et al, 2010). Triple kinase inhibition with IBL-301 initiated profound neurite outgrowth, a morphological sign of differentiation, in PDX LU-NB-3 cells as well as SK-N-BE(2)c and SK-N-SH neuroblastoma cell lines (Figs 3D and EV1D). These results were confirmed by Tuj1 staining and neurite outgrowth quantification (Figs 3D and E, and EV1E). In addition, gene expression of the neuronal differentiation marker GAP43 was slightly upregulated (Fig EV2A), and Gap43 protein expression was induced in LU-NB-3 PDX and SK-N-BE(2)c cells (Fig 3F).

### N-Myc protein expression is downregulated following IBL inhibitor treatment

Amplification of the MYCN oncogene correlates with aggressive neuroblastoma growth, and we thus investigated putative correlations between MYCN and PIM isoform expression levels. There were no differences in PIM1 expression in MYCN-amplified vs. non-amplified tumors, whereas PIM3 was expressed at significantly higher levels in MYCN-amplified tumors (Fig EV2B). We therefore

**Figure 1. Structures of multikinase inhibitors.**

A, B  Chemical structures of IBL-301 (A) and IBL-302 (B).
C  IC50 data of IBL-301, IBL-302, and IBL-202. $IC_{50}$ values were calculated representing the percentage of inhibition against compound concentration and adjusting the experimental data to sigmoidal curve using the software Activity base from IDBS.

investigated whether *PIM1* and *PIM3* had prognostic effects independent of *MYCN* through multivariate analyses. *PIM3* expression did indeed fall out as significant independent prognostic variable in a multivariate cox regression analysis including *MYCN* status ($P = 0.007$), while *PIM1* expression did not ($P = 0.084$; Fig EV2C). IBL-202/301 treatment of PDX cells and neuroblastoma cell lines resulted in unchanged *MYCN* mRNA levels (Fig EV2D) but pronounced decreases in N-Myc protein levels (Fig 3G).

## Multikinase PIM/PI3K/mTOR inhibition induces neuroblastoma cell death

Treatment with IBL-202 and IBL-301 reduced cell viability in two PDX lines and two conventional neuroblastoma cell lines, and the triple PIM/PI3K/mTOR inhibitor IBL-301 had distinctly lower $GI_{50}$ (Fig 4A and Appendix Table S2). To determine whether the decrease in viable cells following IBL treatment was due to cell death and not solely a result of decreased proliferation, we analyzed the cell cycle distribution of neuroblastoma LU-NB-3 PDX cells. The fraction of cells in sub-G1 phase (i.e., non-viable cells) increased after treatment, with the most profound induction by the triple inhibitor IBL-301 (Fig 4B and C). We further showed that the increase in cell death was mediated via apoptosis as assessed by increased cleaved caspase-3 levels (Fig 4D) as well as an increased fraction of Annexin V- and propidium iodide (PI)-positive cells in PDXs and conventional neuroblastoma cell lines (Fig 4E and F).

## IBL-302 reduces neuroblastoma cell viability *in vitro* and tumor growth *in vivo*

IBL-302, a compound with comparable structure to IBL-301, but with increased bioavailability, was chosen for further testing. We first reproduced *in vitro* data and confirmed that IBL-302 inhibited downstream target activity (Fig 5A and B, and Appendix Fig S2A and B). IBL-302 induced neuronal differentiation in PDX as well as two neuroblastoma cell lines as assessed morphologically and by quantification of neurite outgrowth (Fig 5C and D, and Appendix Fig S2C). There was a concentration-dependent reduction in N-Myc protein expression (Fig 5E). In similarity to previous data, we observed reduced neuroblastoma cell viability at nanomolar concentrations (Fig 5F). We could confirm that these changes were due to induced cell death by the increased proportions of Annexin V- and PI-positive cells (Fig 5G).

To assess how these results translated *in vivo*, we xeno-transplanted neuroblastoma cells into the flanks of immunodeficient Nu/Nu mice. Mice were randomly allocated to either control or IBL-302 groups ($n = 5$ in each group) and treated with IBL-302 per orally (p.o.) 5 days a week. IBL-302-treated mice did not display any signs of compound toxicity, and all maintained their body weight throughout treatment (Appendix Fig S2D). Within a few days of starting treatment, there were observable differences in tumor growth, which increased throughout the experiment (Fig 5I). IBL-302 markedly slowed tumor growth (Fig 5I and Appendix Fig S2E) and prolonged survival (Fig 5J).

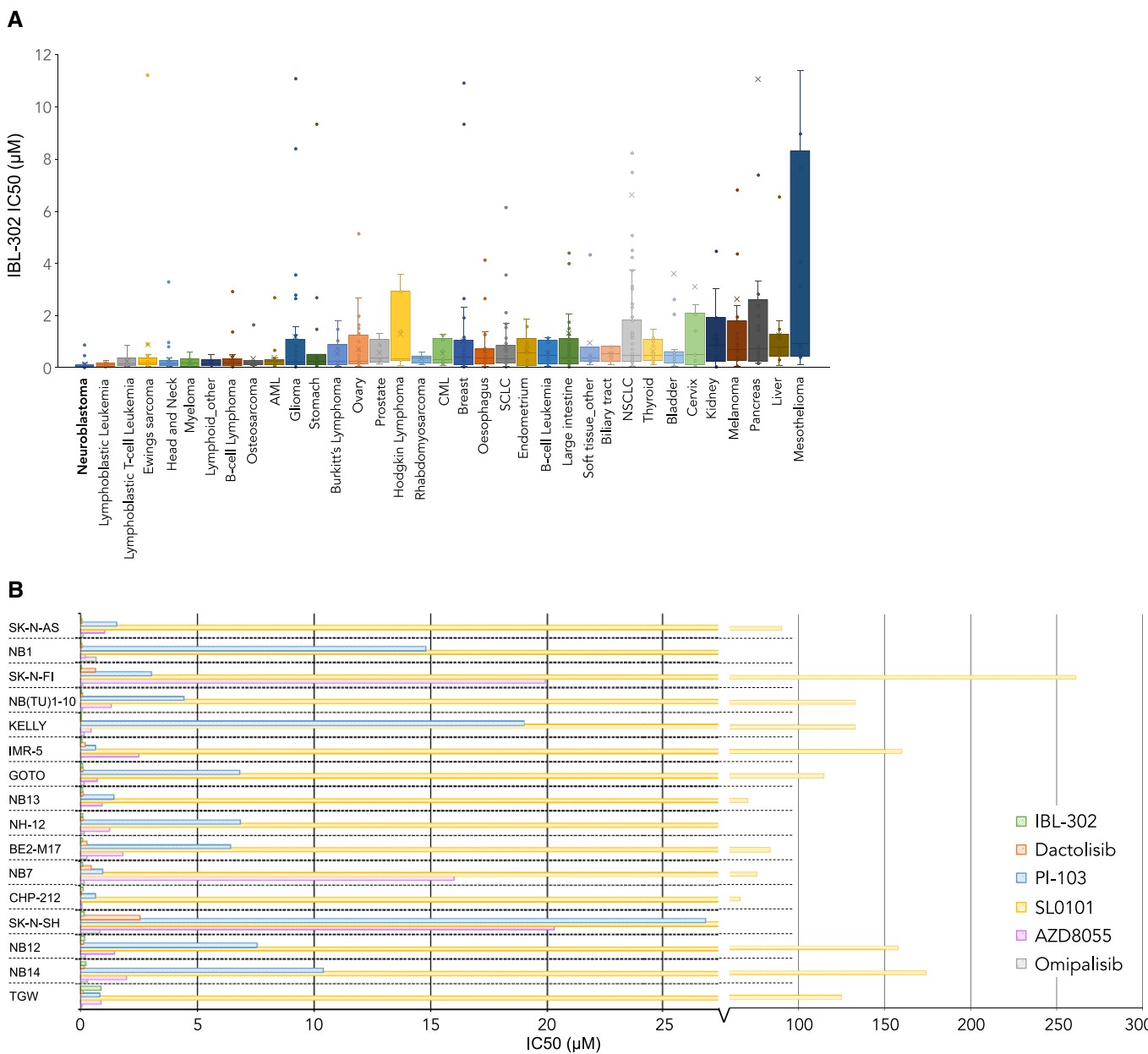

**Figure 2. Neuroblastoma is particularly sensitive to the multikinase inhibitor IBL-302.**

A   Screening of 707 cell lines from 47 tumor types for sensitivity against IBL-302. Tumor forms with $n \geq 5$ cell lines screened are displayed. Comprehensive screening data are available in Appendix Supplementary Materials. Horizontal bands are the median line; the upper part of the box is the 1$^{st}$ quartile and the lower box is the 3$^{rd}$ quartile; the error bars are the maximum and minimum values excluding outliers; the X symbols are mean values. The diagram should show that the y-axis is cut at IC$_{50}$ = 12, for visualization. Outliers range up to IC$_{50}$ ∼ 400.

B   Screening of 16 neuroblastoma cell lines for sensitivity against IBL-302 vs. five PI3K only inhibitors.

## Anti-tumor effects from multikinase targeting

To investigate putative superior effects that triple kinase inhibition holds over single-target inhibition, we set out to analyze the effect of adding PIM and mTOR inhibitors to single inhibition of PI3K. PI3K inhibitor PI-103, mTORC1/2 inhibitor PP242, and PIM inhibitor AZD1208 were used alone or in combination. First, we established EC$_{50}$ values for all three single-target inhibitors in

neuroblastoma PDX cells (Fig EV3A–C), and these EC$_{50}$ values were used for subsequent combination analysis. There was a trend toward increased neuroblastoma cell death following the combination of PI3K, mTORC1/2, and PIM inhibition as compared to single-target inhibition (Fig EV3D and E). We expanded our analysis on the effects of multikinase inhibition by comparing IBL-302 to PI3K inhibitors alone in neuroblastoma cell lines SK-N-AS and SK-N-FI. Both PI3K inhibitors PI-103 and dactolisib induced a

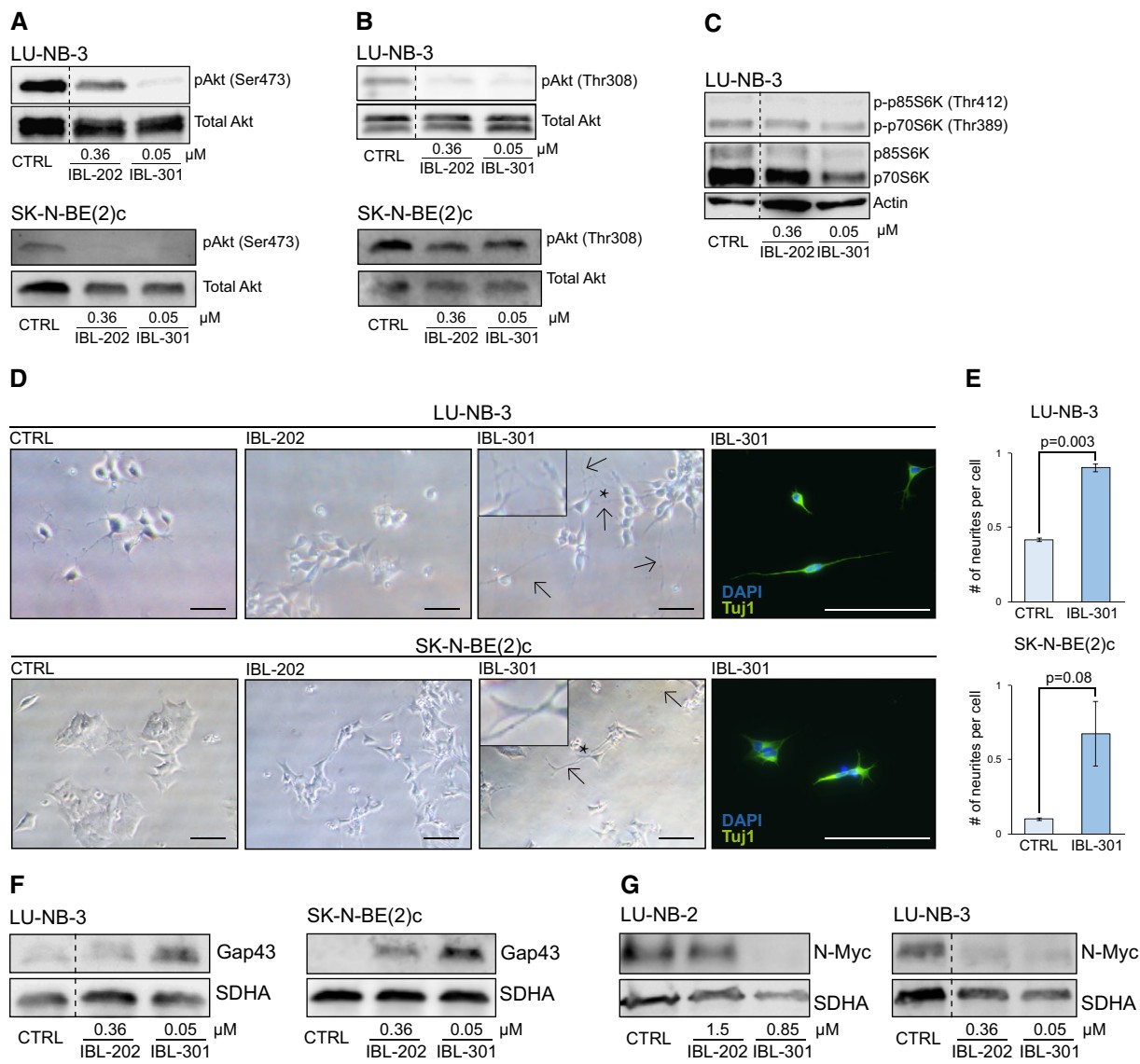

**Figure 3. PIM/PI3K/mTOR inhibition decreases N-Myc levels and increases cellular differentiation.**

Neuroblastoma PDX and SK-N-BE(2)c cells treated with IBL inhibitors at indicated concentrations for 48 h.

A, B    pAkt [at Ser473 (A) and Thr308 (B) sites] levels in LU-NB-3 and SK-N-BE(2)c cells determined by Western blotting. Total Akt levels were used as loading control.

C    p-p70S6K and p-p85S6K levels in LU-NB-3 cells determined by Western blotting. Actin, p70S6K, and p85S6K levels were used as loading controls.

D    Brightfield photomicrographs of LU-NB-3 and SK-N-BE(2)c cells treated with 0.36 μM IBL-202 or 0.05 μM IBL-301. Scale bars represent 100 μm (LU-NB-3) or 200 μm (SK-N-BE(2)c). Arrows indicate neurite outgrowths, and asterisks indicate where inserts are magnified. IBL-301-treated cells were stained for Tuj1. DAPI was used to visualize nuclei.

E    Quantification of neurite outgrowth presented as number of neurites/cell in LU-NB-3 PDX and SK-N-BE(2) cells treated with IBL-301. For LU-NB-3 PDX cells, representative areas (*n* = 2) were used and *n* = 344 and *n* = 240 cells/condition for CTRL and IBL-301, respectively, were counted. For SK-N-BE(2)c cells, representative areas (*n* = 2 and *n* = 3 for CTRL and IBL-301, respectively) were used and *n* = 141 and *n* = 130 cells/condition for CTRL and IBL-301, respectively, were counted. Values are reported as mean ± SEM. Statistical significance was determined by two-sided Student's *t*-test. *P* = 0.003 for LU-NB-3 and *P* = 0.08 for SK-N-BE(2)c.

F    Gap43 protein levels in LU-NB-3 and SK-N-BE(2)c cells determined by Western blotting. SDHA levels were used as loading control.

G    N-Myc levels in LU-NB-2 and LU-NB-3 cells determined by Western blotting. SDHA levels were used as loading control.

Source data are available online for this figure.

dose-dependent reduction in cell viability, although not as efficiently as IBL-302 (Fig EV3F). We did however not observe any significant differences in tumor growth *in vivo* between IBL-302 and dactolisib treatment when xeno-transplanting SK-N-AS cells into Nu/Nu mice (Fig EV3G).

**PIM/PI3K/mTOR inhibition potentiates the effect of conventional chemotherapy**

The majority of young neuroblastoma survivors experience later life effects from treatment with chemotherapy (Norsker *et al*, 2018).

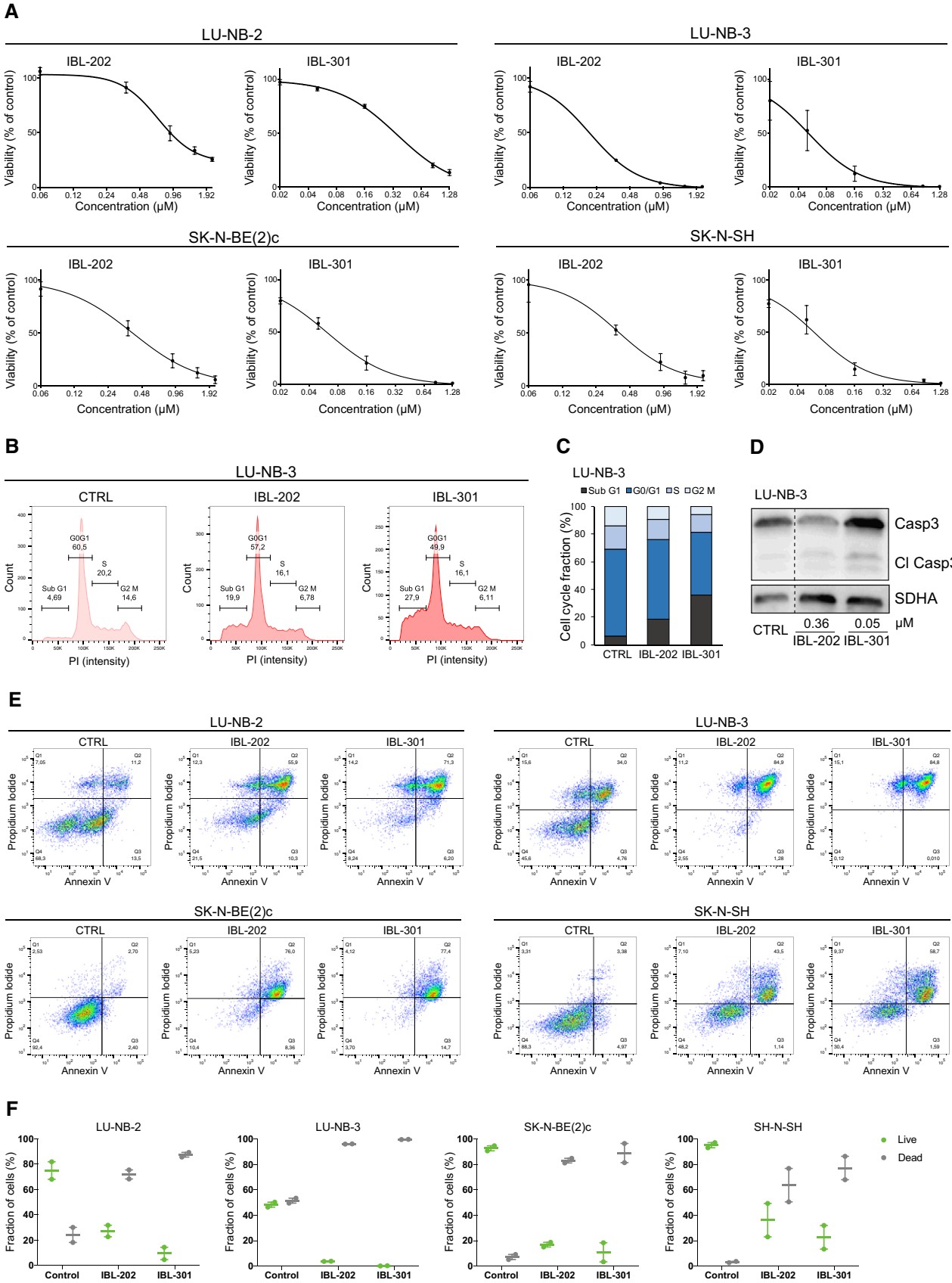

**Figure 4.**

◀

**Figure 4.  Treatment with multikinase inhibitors results in cell death.**

A       Viability of IBL-202- or IBL-301-treated cells determined by CellTiter-Glo. Values are reported as mean ± SEM. $n$ = 2 replicates for LU-NB-2 and LU-NB-3, $n$ = 3 replicates for SK-N-BE(2)c and SK-N-SH.

B, C    Cell cycle distribution of LU-NB-3 cells determined by flow cytometry. Bars show mean values from two independent experiments.

D       Cleaved caspase-3 (Cl Casp3) levels in LU-NB-3 cells determined by Western blotting. SDHA levels were used as loading control.

E       Flow cytometry analyses of Annexin V and PI stainings following treatment with 360 nM IBL-202 or 50 nM IBL-301.

F       Quantification of live and dead cells from the Annexin V/PI stainings. Dead cells = PI positive, live cells = PI negative. Dot plots from two independent experiments and error bars represent SEM.

Source data are available online for this figure.

Adding IBL-202 or IBL-301 to cisplatin, a component of current neuroblastoma therapy, demonstrated synergistic effects with regard to cell viability (Fig 6A). IBL-302 showed an additive effect with doxorubicin and etoposide, two further chemotherapies used in neuroblastoma patients (Appendix Fig S3A and B). In the case of cisplatin, concentrations of this chemotherapeutic could be reduced by ~50% when combined with low-dose IBL-302, with retained negative effects on cell viability (Fig 6B). Of note, IBL-302 and cisplatin acted synergistically in reducing cell viability in LU-NB-3 PDX and SK-N-BE(2)c cells (Fig 6B). Possible synergistic effects of IBL-302 and cisplatin are however not as clear in the SK-N-SH cells, displaying additive as well as antagonistic effects depending on the cisplatin concentration (Fig 6B). One possible explanation is that LU-NB-3 and SK-N-BE(2)c cells are *MYCN*-amplified, while SK-N-SH is a non-*MYCN*-amplified cell line. In line with the inhibitory effects observed on N-Myc protein levels following IBL-302 treatment, PIM/PI3K targeting might have a stronger effect against *MYCN*-amplified neuroblastoma as compared to non-*MYCN*-amplified tumors. As single-line treatments, IBL-302 and cisplatin slightly increased the amount of neuroblastoma cells in the sub-G1 phase of the cell cycle and when these two compounds were combined the fraction of sub-G1 phase cells increased further (Fig 6C and D).

**Triple kinase inhibitor IBL-302 potentiates the effects of cisplatin in PDX models *in vivo***

To assess whether the effect of IBL-302 on cisplatin treatment *in vitro* was also true *in vivo*, we xeno-transplanted LU-NB-3 PDX cells into immunodeficient Nu/Nu mice and examined the growth-inhibiting effect of combined IBL-302/cisplatin treatment. Mice were allocated randomly to either control, IBL-302 alone, cisplatin alone, or IBL-302 and cisplatin combination groups ($n$ = 5 in each group). Aiming to reduce the concentration of otherwise toxic cisplatin, we treated mice with low-dose cisplatin as well as lower doses (50% of maximum tolerated dose) of IBL-302. Again, treatment did not induce any overt side effects in terms of either general mouse health or body weight (Appendix Fig S3C).

There were no major differences in tumor growth between control and low-dose IBL-302 (Fig 6E and F). In cisplatin-treated mice, there were two response patterns: One group did not respond, whereas the other partially responded in terms of reduced tumor growth (Fig 6E and F). No mice in either of control, IBL-302 alone, or cisplatin alone groups survived the study period (70 days; Fig 6E–G). However, in the IBL-302/cisplatin combination group, there was a subgroup of partial responders, similar to cisplatin alone, and two mice that substantially responded to treatment, displaying slow or no growth and surviving the study period (Fig 6E–G). There was a significant difference in survival between the four treatment groups ($P$ = 0.0014; Fig 6G).

Tumor sections from all four study groups were stained for the expression of N-Myc protein. As expected, all tumors were N-Myc-positive, but, of note, there was a decrease in nuclear N-Myc expression in tumors from the combination group (Fig 6H).

**Global gene and protein analyses identify important IBL-302-regulated biological processes**

Since IBL-302 demonstrated effects on neuroblastoma PDX cells *in vitro* and *in vivo*, we next investigated which biological processes mediated these effects by treating PDX cells with nanomolar concentrations of IBL-302 and consequently performed RNAseq and mass spectrometry analyses. Several cellular processes were influenced by IBL-302, including cell motility, apoptosis, programmed cell death (e.g., *CASP3*), and cell cycle (e.g., *CDK6* and *CCND1*; Fig 7A–F and Dataset EV1). Consistent with these data, caspase-3 and CDK6 were among the most significantly differentially expressed proteins, being upregulated by IBL-302 treatment as detected by mass spectrometry (Fig 7E and F, and Dataset EV1).

To understand the relationship between IBL-302-induced RNA and protein changes, we compared global gene and protein expression in each individual sample (control $n$ = 4; IBL-302 $n$ = 2). RNA and protein expression levels were well-correlated ($R$ > 0.50 in all samples), suggesting that IBL-302 treatment significantly affects both transcriptional and translational associated processes in neuroblastoma (Figs 7G and EV4A). While correlations between phosphorylated proteins and RNA expression were also robust ($R$ > 0.39 in all samples; Fig EV4B), the association between protein and phosphorylated protein data was weaker ($R$ ≤ 0.29; Fig EV4C).

Importantly, proteome (Fig 7E) and phospho-proteome (Fig 7F) data identified several of the biological processes retrieved from the RNAseq analysis (Fig 7D). Taken together, the RNA, protein, and phospho-protein data show that the signaling of PIM, PI3K, and mTOR seems to be of importance for regulating neuroblastoma cell apoptosis.

# Discussion

Constitutive activation or other alterations in the PI3K pathway are certainly some of the most common aberrations in cancer (Engelman, 2009). Consequently, several PI3K inhibitors have reached clinical trials and even the clinic (Fruman *et al*, 2017). Although promising, some patients develop treatment resistance, in many cases due to activation of parallel pathways that circumvent PI3K inhibition and maintain tumor cell growth advantages. Activation of the PIM kinases is thought to be an important PI3K inhibitor resistance mechanism (Le *et al*, 2016). While little is known about the

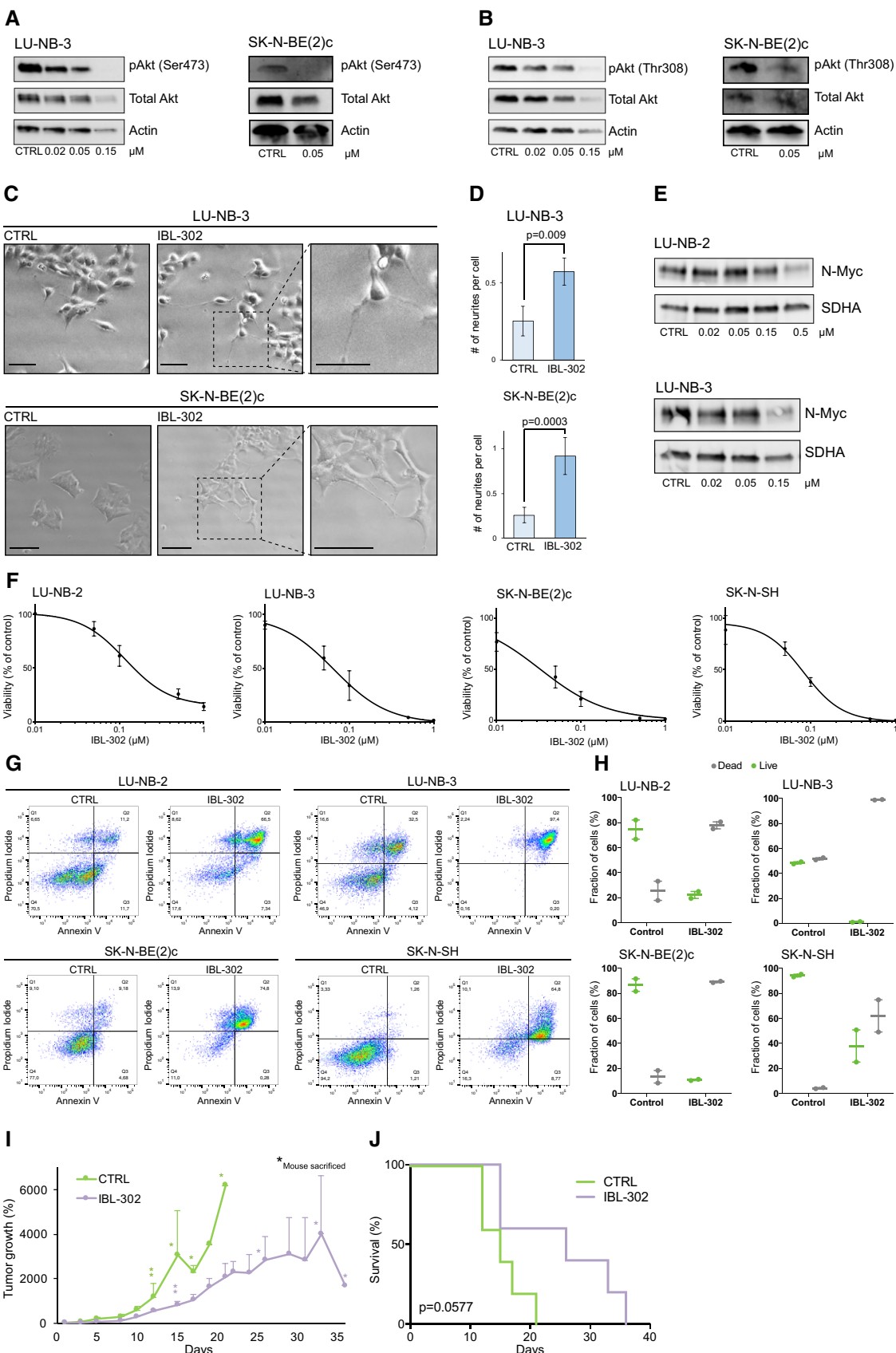

**Figure 5.**

**Figure 5.  IBL-302 reduces neuroblastoma growth in vivo.**

Neuroblastoma cells were treated with indicated concentrations of IBL-302 for 48 h (pAkt and N-Myc expression, photomicrographs, and flow cytometry) or 72 h (cell viability, (F)).

A   Expression of pAkt(Ser473) in LU-NB-3 and SK-N-BE(2)c cells. Total Akt and actin were used as loading controls.
B   Expression of pAkt(Thr308) in LU-NB-3 and SK-N-BE(2)c cells. Total Akt and actin were used as loading controls.
C   Brightfield photomicrographs of LU-NB-3 and SK-N-BE(2)c cells treated with 50 nM IBL-302. Scale bars represent 100 μm (LU-NB-3) or 200 μm (SK-N-BE(2)c).
D   Quantification of neurite outgrowth presented as number of neurites/cell in LU-NB-3 PDX and SK-N-BE(2)c cells treated with IBL-302. For LU-NB-3 PDX cells, representative areas ($n = 4$) were used and $n = 460$ and $n = 216$ cells/condition for CTRL and IBL-302, respectively, were counted. For SK-N-BE(2)c cells, representative areas ($n = 5$ and $n = 7$ for CTRL and IBL-302, respectively) were used and $n = 124$ and $n = 260$ cells/condition for CTRL and IBL-301, respectively, were counted. Values are reported as mean ± SEM. Statistical significance was determined by two-sided Student's *t*-test. $P = 0.009$ for LU-NB-3 and $P = 0.0003$ for SK-N-BE(2)c.
E   N-Myc protein expression determined by Western blotting. SDHA was used as loading control.
F   Cell viability determined by CellTiter-Glo. Values are reported as mean ± SEM. $n = 3$.
G   Flow cytometry analyses of Annexin V and PI stainings following treatment with 50 nM IBL-302.
H   Quantification of live and dead cells. Dead cells = PI positive, live cells = PI negative. Dot plots from two independent experiments and error bars represent SEM.
I   Neuroblastoma SK-N-BE(2)c carrying mice ($n = 5$ in each group) were treated with vehicle (CTRL) or 40 mg/kg IBL-302 for up to 35 days, and tumor growth was followed over time. Asterisks indicate each occasion a mouse within that particular group was sacrificed. Values are reported as mean ± SEM.
J   Kaplan–Meier survival curves comparing mice treated with vehicle (CTRL) or IBL-302. Log-rank test was used to determine statistical significance. $P = 0.0577$. $n = 5$ in each group.

Source data are available online for this figure.

role of PIM kinases in neuroblastoma, a recent report suggested that PIMs might serve as prognostic biomarkers and putative treatment targets in this tumor type (Brunen *et al*, 2018).

Here, we investigated the use of novel multikinase PIM/PI3K/mTOR inhibitors in neuroblastoma treatment. Screening hundreds of cell lines from nearly 50 tumor types confirmed that neuroblastoma was highly sensitive to triple kinase inhibition. We show that the PIM/PI3K/mTOR inhibitors IBL-301 and IBL-302 induced neuroblastoma cell differentiation and cell death at nanomolar concentrations. Neuroblastomas presenting with a more differentiated phenotype (i.e., high tumor cell expression of sympathetic neuronal markers) are associated with less aggressive growth and a lower risk of metastasis (Fredlund *et al*, 2008), and therapy that induces sympathetic neuronal differentiation is a promising treatment strategy. Triple PIM/PI3K/mTOR inhibition induced differentiation of neuroblastoma cells, increasing the expression of the sympathetic neuronal marker Gap43 and promoting neurite outgrowth.

Amplification of the *MYCN* oncogene is present in ~25% of all neuroblastomas and is a marker for aggressive tumors, and in line with a differentiated phenotype, we detected lower N-Myc expression in our PDX cells following IBL-302 treatment. These findings are consistent with previous reports on PI3K inhibition in neuroblastoma (Chesler *et al*, 2006; Johnsen *et al*, 2008; Chanthery *et al*,

2012; Cage *et al*, 2015). Similarly, PIM inhibition has previously been suggested as a therapeutic option in MYC/MYCN-associated tumor types (Kirschner *et al*, 2014; Horiuchi *et al*, 2016; Brunen *et al*, 2018). Our results of combined PIM/PI3K inhibition effects in neuroblastoma are in line with findings from other tumor types including breast cancer (Le *et al*, 2016) and glioblastoma (Iqbal *et al*, 2016). Future studies should address the design of optimal drug combinations using IBL-302 with other targeted therapy and/or chemotherapy for high-risk neuroblastoma.

Transcriptomic and proteomic analyses following PIM/PI3K/mTOR inhibition confirmed our *in vitro* results and suggested that IBL-302 treatment induces apoptosis, with caspase-3 being one of the most differentially expressed genes/proteins as detected by both RNA sequencing and mass spectrometry. In addition, cell cycle regulation, differentiation, and cell migration processes were dysregulated following IBL-302 treatment. Protein and mRNA expression strongly correlated in our datasets, suggesting that IBL-302 affects both transcriptional and translational processes in neuroblastoma.

Emerging data suggest that high-risk neuroblastomas harbor a high degree of intratumor heterogeneity and contain tumor subclones with distinct genetic/epigenetic and/or phenotypic characteristics (Eleveld *et al*, 2015; Mengelbier *et al*, 2015; Schramm *et al*, 2015; Boeva *et al*, 2017; van Groningen *et al*, 2017; Braekeveldt *et al*, 2018; Karlsson *et al*, 2018). These findings suggest that

**Figure 6.  PIM/PI3K/mTOR inhibition enhances the effects of chemotherapy.**

A, B   LU-NB-3 PDX, SK-N-BE(2)c, and SK-N-SH cells were treated with indicated concentrations of cisplatin and/or 0.36 μM IBL-202, 0.05 μM IBL-301, or 0.05 μM IBL-302 for 48 h. Cell viability determined by CellTiter-Glo. Graphs show mean values and SEM from three independent experiments. Theoretical additive curves were evaluated by calculating the combination index (CI) based on the Bliss independence model.
C, D   Cell cycle distribution in LU-NB-3 PDX cells after treatment with 320 nM cisplatin, 50 nM IBL-302, or the combination of these, determined by flow cytometry. Bars show mean values from two independent experiments.
E   Tumor size in individual mice from (F.) presented as each group separately.
F   Neuroblastoma PDX carrying mice ($n = 5$ in each group) were treated with vehicle (CTRL), low-dose (20 mg/kg) IBL-302, low-dose (1 mg/kg) cisplatin, or the combination of low-dose IBL-302 and low-dose cisplatin for up to 70 days. Tumor size was measured over time. Asterisks indicate each occasion a mouse within that particular group was sacrificed. Values are reported as mean ± SEM.
G   Kaplan–Meier survival curves comparing mice treated with vehicle (CTRL), IBL-302, cisplatin, or the combination. Log-rank test was used to determine statistical significance. $P = 0.0014$. $n = 5$ per group.
H   Immunohistochemical staining of N-Myc expression in tumors from (F). Scale bars represent 100 μm. Roman numbers in each box indicate which mouse the tumor derives from (see individual graphs in E).

Source data are available online for this figure.

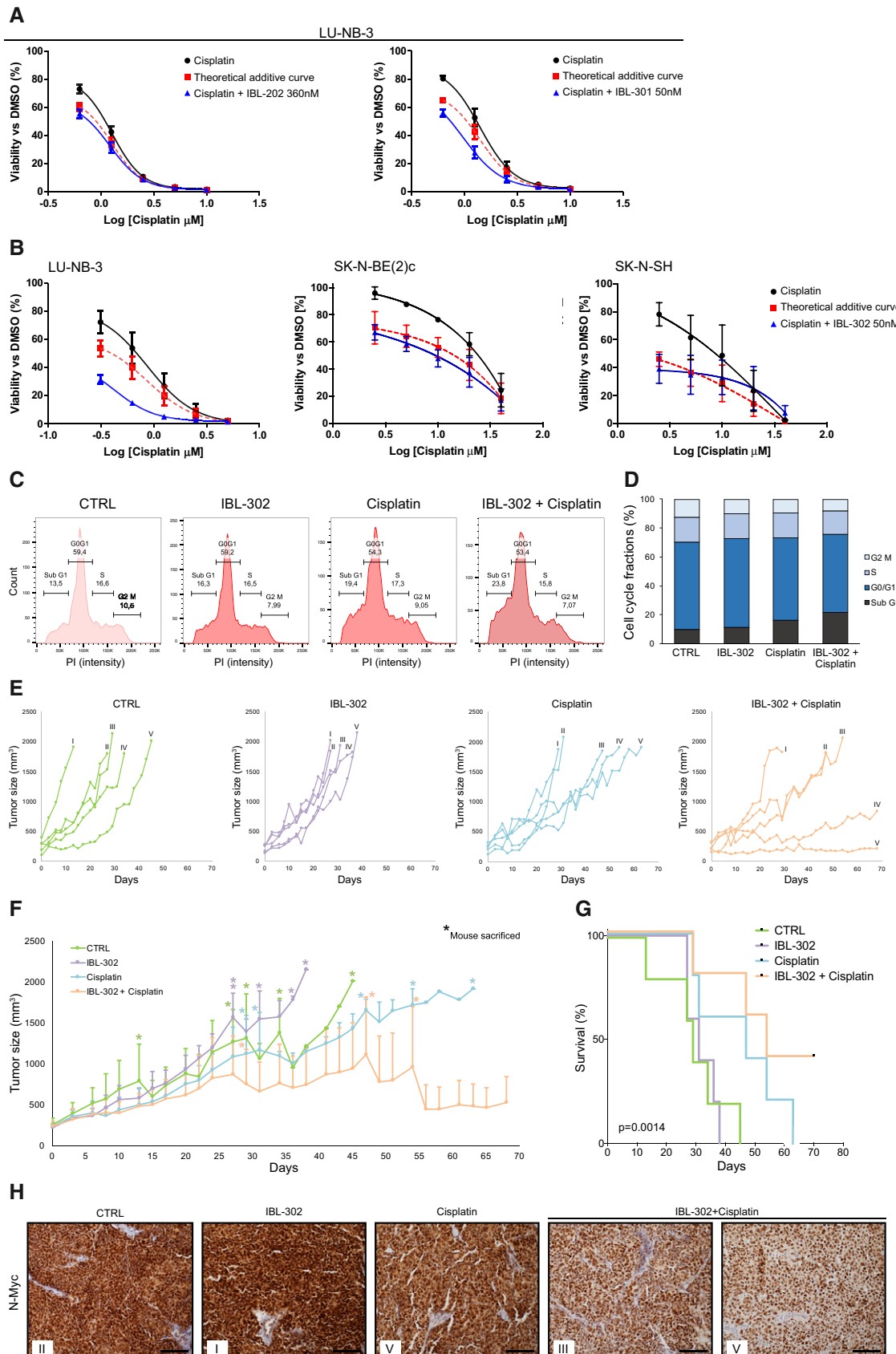

**Figure 6.**

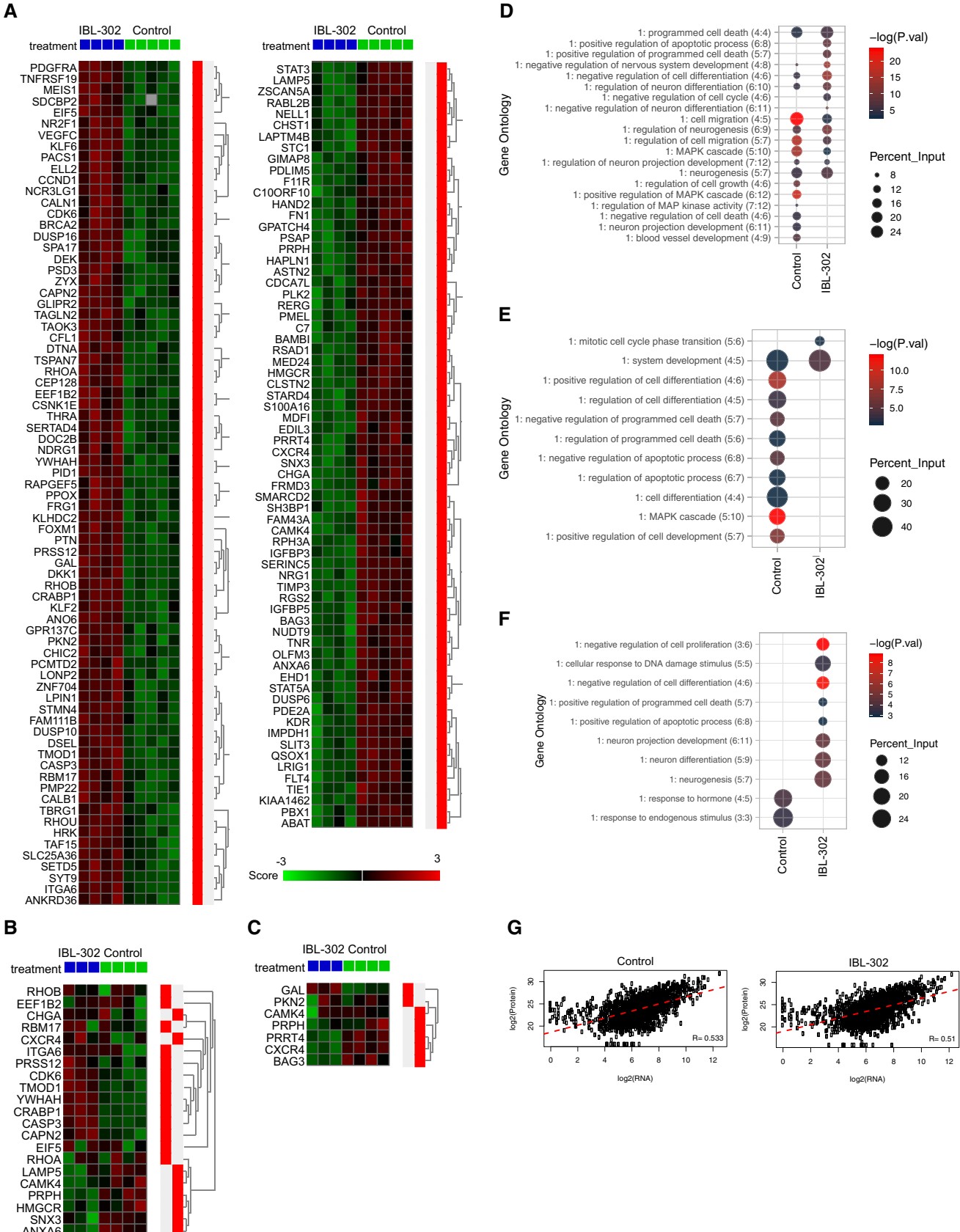

**Figure 7.**

◄

**Figure 7. Tumor-related processes are affected by IBL-302 treatment.**

LU-NB-3 PDX cells were treated with 50 nM of IBL-302 for 48 h and subsequently analyzed by RNAseq and mass spectrometry.

A–C  Heat map of significantly differentially expressed genes across treatment groups from RNAseq data (FDR < 0.1) (A). Genes defined as differentially expressed that were also present in MS and phospho-MS are visualized in (B) and (C), respectively.

D  Gene ontology generated from significantly altered genes from RNAseq data. −log(P-values) and percent of input gene/protein lists that overlap with the specified ontology are displayed.

E  Gene ontology generated from significantly altered proteins from mass spectrometry proteome data. −log(P-values) and percent of input gene/protein lists that overlap with the specified ontology are displayed.

F  Gene ontology generated from significantly altered phospho-proteins from mass spectrometry phospho-proteome data. −log(P-values) and percent of input gene/protein lists that overlap with the specified ontology are displayed.

G  Cross-platform sample correlations of RNAseq and mass spectrometry proteome data. Pearson's correlation coefficients are displayed.

general and/or combinatorial therapeutic strategies must be further explored. Our combined PIM/PI3K/mTOR inhibition is a viable treatment strategy for neuroblastoma. We also show that the combination of low-dose IBL-302 (50% of maximum tolerated dose) and low-dose cisplatin suppressed neuroblastoma growth in PDX *in vivo* models. These findings represent a novel promising approach for children with high-risk *MYCN*-amplified neuroblastoma, in terms of not only target but also dosing strategy and companion diagnostic.

## Materials and Methods

### PI3 kinase assay

The kinase activity of PI3K alpha was measured by using the commercial ADP Hunter™ Plus assay (DiscoveRx), a homogeneous assay measuring ADP accumulation, as a universal product of kinase activity. The assay was done following general manufacturer recommendations and adapting protein and substrate concentrations to optimal conditions. Kinase buffer was 50 mM HEPES, pH 7.5, 100 mM NaCl, 1 mM EGTA, 0.04% CHAPS, 3 mM $MgCl_2$, and 10 μg/ml BGG (bovine γ-globulin), 2 mM TCEP. The assay was done at 100 μM PIP2, as peptide substrate, and 50 μM ATP. Protein concentration was 0.8 ng/μl. In order to calculate the $IC_{50}$ of described compounds, serial 1:5 dilutions were prepared, and the reaction started by the addition of ATP. Incubation was done for 1 h at 25°C. Reagents A and B (DiscoveRx) were sequentially added to the wells, and plates were incubated for 30 min at 37°C. Fluorescence counts were read in a Victor instrument (PerkinElmer) with the recommended settings (544 and 580 nm as excitation and emission wavelengths, respectively). Values were plotted against inhibitor concentration and fit to a sigmoid dose–response curve with the GraphPad software.

### mTOR biochemical assay

The biochemical assay to measure mTOR activity relies on the LanthaScreen™ kinase activity assay (Invitrogen), where kinase, GFP-labeled substrate, and ATP are allowed to react. EDTA (to stop the reaction) and terbium-labeled antibody (to detect phosphorylated product) are then added. In a LanthaScreen™ kinase reaction, the antibody associates with the phosphorylated GFP-labeled substrate resulting in an increased TR-FRET value. The TR-FRET value is a dimensionless number that is calculated as the ratio of the acceptor (GFP) signal to the donor (terbium) signal. The amount of

antibody that is bound to the tracer is directly proportional to the amount of phosphorylated substrate present, and in this manner, kinase activity can be detected and measured by an increase in the TR-FRET value.

The mTOR enzyme has been purchased from Invitrogen mTOR, as well as the GFP-labeled substrate (4EBP1-GFP) and the Tb-anti-p4EBP1 (pThr46) antibody. Assay conditions were as indicated by the kit manufacturers. Assays were performed in 96-well plates. The final read out was generated using an EnVision plate reader (PerkinElmer) The TR-FRET value (a dimensionless number) was calculated as the ratio of the acceptor signal (GFP, emission at 520 nm) to the donor signal (terbium, emission at 495 nm). Values were plotted against the inhibitor concentration and fitted to a sigmoid dose–response curve using GraphPad software.

### PIM kinase assays

The PIM kinase activities were measured using the commercial ADP Hunter™ Plus assay (DiscoveRx), a homogeneous assay measuring ADP accumulation, as a universal product of kinase activity. The assay was done following general manufacturer recommendations and adapting protein and substrate concentrations to optimal conditions. Kinase buffer was 15 mM HEPES, pH 7.4, 20 mM NaCl, 1 mM EGTA, 0.02% Tween-20, 10 mM $MgCl_2$, and 0.1 mg/ml LBGG (bovine γ-globulin). All PIM kinase assays were done at 100 μM PIMtide (ARKRRRHPSGPPTA), as peptide substrate, and 100 μM ATP. Protein concentration was 50, 350, and 500 ng/μl for PIM1, PIM2, and PIM3, respectively. In order to calculate the $IC_{50}$ of described compounds, serial 1:5 dilutions were prepared, and the reaction started by the addition of ATP. Incubation was done for 1 h at 25°C. Reagents A and B (DiscoveRx) were sequentially added to the wells, and plates were incubated for 30 min at 37°C. Fluorescence counts were read in a Victor instrument (PerkinElmer) with the recommended settings (544 and 580 nm as excitation and emission wavelengths, respectively). Values were plotted against inhibitor concentration and fit to a sigmoid dose–response curve with the GraphPad software.

### Patient samples

PDX samples were previously established and have been described in detail (Braekeveldt *et al*, 2015; Persson *et al*, 2017). LU-NB-2 cells were derived from a brain metastasis following chemotherapy relapse (Stage IV; patient age 2y2m). LU-NB-3 cells were derived from an untreated primary tumor in the adrenal gland (Stage III; patient age 2y9m). LU-NB-2 and LU-NB-3 were

cultured as described previously (Persson *et al*, 2017). Briefly, PDX-derived cells were maintained in a mix of Dulbecco's modified Eagle's medium (DMEM) and GlutaMAX™ F-12 (3:1 ratio) supplemented with 1% penicillin/streptomycin (P/S), 2% B27 w/o vitamin A, 40 ng/ml basic fibroblast growth factor (FGF), and 20 ng/ml epidermal growth factor (EGF), with no addition of FBS. For adherent growth purposes, PDX-derived cells were grown on laminin-coated wells (10 μg/ml laminin-521, Biolamina). All cells in culture were regularly replaced and screened for mycoplasma. Cell authentication was performed through SNP profiling of PDX-derived cell lines and their corresponding PDXs (Multiplexion, Heidelberg, Germany).

## Cell lines

The human SK-N-BE(2)c (CRL-2268; a kind gift from Dr. June Biedler), SK-N-SH, SK-N-FI, and SK-N-AS cell lines were acquired from ATCC. Neuroblastoma cells were cultured in minimal essential medium (MEM; SK-N-BE(2)c and SK-N-SH) or RPMI (SK-N-FI and SK-N-AS) supplemented with fetal bovine serum (FBS) and P/S. All cells in culture were regularly replaced and screened for mycoplasma. Cell authentication was performed through SNP profiling (Multiplexion, Heidelberg, Germany).

## Screening

Publicly available data on GI$_{50}$ values for omipalisib, AZD8055, SL0101, PI-103, and dactolisib across a panel of 16 neuroblastoma cell lines were obtained from the Genomics of Drug Sensitivity in Cancer (GDSC) database (Yang *et al*, 2013; Release 7.0). GI$_{50}$ data for IBL-302 across 707 cell lines from 47 tumor types and across a separate panel of 16 neuroblastoma cell lines were obtained from the GDSC consortium (Yang *et al*, 2013).

## Reagents

Multikinase inhibitors IBL-202 (PIM/PI3K), IBL-301 (PIM/PI3K/ mTOR), and IBL-302 (PIM/PI3K/mTOR) were reconstituted in 100% DMSO for *in vitro* experiments and used at concentrations surrounding the GI$_{50}$ range for respective drug (0–2.16 μM for IBL-202; 0–1.28 μM for IBL-301; 0–1.0 μM for IBL-302). When fixed concentrations of the IBL inhibitors were used *in vitro*, these were chosen based on viability assays as lowest possible dose but with viable effects. For *in vivo* experiments, IBL-302 was reconstituted in either DMSO:PEG400 (10:90 v/v) or DMSO: PEG400:vitamin E TPGS (10:70:20 v/v) and administered at 40 or 20 mg/kg p.o five times a week. Chemotherapeutic drugs cisplatin, doxorubicin, and etoposide were diluted in sterile H$_2$O and used at concentrations as indicated in *in vitro* experiments (0–5 μM for cisplatin; 0–50 nM for doxorubicin; 0–160 nM for etoposide). For *in vivo* treatment, cisplatin was reconstituted in sterile saline and administered at 1 mg/kg intraperitoneally (i.p) three times a week. Vehicle controls were DMSO for *in vitro* experiments and DMSO:PEG400 or DMSO:PEG400:vitamin E TPGS for *in vivo* experiments. Inhibitors dactolisib, PI-103, AZD1208, and PP242 were diluted in 100% DMSO for *in vitro* work, and dactolisib was diluted in 10% DMSO and 90% PEG400 for *in vivo* work.

## RNA extraction and qRT–PCR

Total RNA was automatically extracted using the Arrow with Arrow RNA (Tissue Kit-DNA Free) Kit (DiaSorin, Saluggia, Italy) and converted to cDNA using the MultiScribe Reverse Transcriptase Enzyme (Applied Biosystems, Foster City, CA) with random primers. Quantitative RT–PCR was performed with the SYBR Green PCR Master Mix (Life Technologies, Carlsbad, CA) using the comparative Ct method (Vandesompele *et al*, 2002). Three reference genes (*YWHAZ*, *UBC*, and *SDHA*) were used to normalize gene-of-interest expression. Primer sequences can be found in Appendix Table S3.

## Western blot

Cells were lysed in RIPA buffer supplemented with Complete Protease Inhibitor (Roche, Basel, Switzerland) and phosSTOP (Roche) as previously described (Mohlin *et al*, 2015). Proteins were separated on SDS–PAGE gels and transferred to PVDF- or Hybond-C extra nitrocellulose membranes (Bio-Rad, Hercules, CA). Antibodies are listed in Appendix Table S4.

## Immunofluorescence

Adherent SK-N-BE(2)c cells were cultured and treated on presterilized cover slips, whereas PDX-derived cells were grown on precoated laminin-covered (10 μg/ml laminin-521, Biolamina) glass slides. All cells were fixed in 4% paraformaldehyde before antibody blocking and cell membrane permeabilization in 5% goat serum and 0.5% Triton-X. Staining of peripheral neurons was performed using the mouse monoclonal anti-human Tuj1 antibody [Covance (MMS-435P), Princeton, NJ; 1:500], and positive cells were visualized by Alexa Fluor 488 goat anti-mouse antibody [Invitrogen (A11029), 1:200]. Cell nuclei were visualized by DAPI staining [Invitrogen (D3571), 1:3,000].

## Cell viability assay

Cells were seeded in triplicates in white opaque 96-well plates, 5,000 cells per well. SK-N-BE(2)c, SK-N-SH, SK-N-AS, and SK-N-FI cells were incubated for 24 h before treatment, while the PDX-derived cells LU-NB-2 and LU-NB-3 were treated directly after seeding. Treated cells were incubated for 48 or 72 h, and cell viability was measured using the CellTiter-Glo assay (G7571; Promega, Madison, WI).

## Annexin V

Cell death was evaluated through Annexin V and PI staining. Treated cells were stained with 5% APC Annexin V and 2% propidium iodide (PI). The samples were incubated for 15 min and then run on a FACSVerse flow cytometer (BD Biosciences). The data were analyzed using the FlowJo software.

## Cell cycle distribution

Cells were treated for 48 h and then fixed in ice-cold 70% ethanol. Samples were kept at −20°C and upon analysis washed in PBS and

incubated on ice for 45 min in Vindelöv solution [3.5 μmol/l Tris–HCl (pH 7.6), 10 mmol/l NaCl, 50 μg/ml propidium iodide, 20 μg/ml RNase, 0.1% v/v NP40]. Samples were run on a FACSVerse flow cytometer (BD Biosciences, Franklin Lakes, NJ), and the data were analyzed using the FlowJo software.

## Chemotherapy combination experiments

Cells were seeded in triplicates in white opaque 96-well plates, 5,000 cells per well. Cells were incubated for 24 h before being treated for 48 h using log-scale concentrations of chemotherapeutic in addition to fixed concentrations of IBL compounds as indicated. Cell viability was measured with CellTiter-Glo as previously described, and data were normalized to the average of all treated wells with DMSO. All synergy experiments were performed as technical triplicates. Potential synergy between chemotherapeutics and IBL compounds was evaluated by calculating the combination index (CI) based on the Bliss independence model (Foucquier & Guedj, 2015), whereby the CI was calculated with the following equation: $CI = \frac{Ea+Eb-Ea*Eb}{Eab}$, where Ea indicates the viability effect of drug A (chemotherapeutic), Eb indicates the viability effect of drug B (IBL compound), and Eab indicates the viability effect of the drug combination. $CI < 1$ indicates synergism, $CI = 1$ indicates additivity, and $CI > 1$ indicates antagonism.

## In vivo experiments

Four- to eight-week-old female athymic mice (NMRI-Nu/Nu) were purchased from Taconic. The regional ethics committee approved all procedures (ethical permits M11-15 and M146-13) and housing facilities comply with official regulations. Animals' general health conditions were checked daily by professional caretakers. Female mice were kept in groups of up to five in standard IVC cages containing bedding and nesting material. Mice were under controlled lighting conditions (12-h light cycles), relative humidity and temperature, food, and water provided *ad libitum*. All efforts were made to minimize animal suffering and reduce the number of animals sacrificed. One million SK-N-BE(2)c or LU-NB-3 cells were diluted in culture medium:matrigel solution (2:1) and injected subcutaneously into the right flank of the mice. Tumor volume was measured three times a week using a digital caliper and calculated with the formula $V = \pi ls^2/6$ (l = long side; s = short side). For the SK-N-BE(2)c study, mice were randomly allocated into one of two groups (control or IBL-302, respectively, $n = 5$ in each group) 11 days after tumor cell injection (average tumor size in each group ~110 mm$^3$). Mice were treated p.o. 5 days a week with 40 mg/kg IBL-302 diluted in 10% DMSO and 90% PEG-400 or the corresponding amount of vehicle. Mice were treated until individual tumor sizes had reached ~1,800 mm$^3$. For the LU-NB-3 PDX study, mice were allocated individually when their respective tumors had reached a minimum size of 100 mm$^3$ (average size in each group ~240 mm$^3$) into one of four groups: control, IBL-302, cisplatin, or IBL-302 and cisplatin combination ($n = 5$ in each group). Mice were treated p.o. 5 days a week with 20 mg/kg IBL-302 diluted in 10% DMSO, 20% vitamin E, and 70% PEG400 and/or i.p. 3 days a week with 1 mg/kg cisplatin diluted in saline. Corresponding amounts of 10% DMSO,

20% vitamin E, and 70% PEG400 served as vehicle control for all groups. Mice were treated until their individual tumor size had reached ~1,800 mm$^3$ or after a maximum of 50 injection days.

For the IBL-302 and dactolisib comparison study, three million SK-N-AS cells were injected subcutaneously and after 10 days mice were allocated to one of three groups: control DMSO/PEG400 10/90), dactolisib (45 mg/ml), or IBL-302 (40 mg/kg). All mice were treated 5 days per week (Monday to Friday) using i.p. injections. Mice were euthanized when their tumor size had reached ~1,800 mm$^3$ or at the end of the study.

## Immunohistochemistry

Dissected tumors from *in vivo* studies were fixed in 4% paraformaldehyde and embedded in paraffin. After antigen retrieval using PT Link (Dako; Agilent Technologies, Santa Clara, CA), sections (4 μm) were stained using the Autostainer Plus (Dako). Antibodies used can be found in Appendix Table S4.

## RNA sequencing

LU-NB-3 cells were seeded on laminin precoated plates (10 μg/ml laminin-521, Biolamina) and allowed to attach for 24 h. Cells were treated with 50 nM IBL-302 or vehicle control for 48 h before harvesting. Each sample consisted of pooled cells from plates treated equally, and samples were then divided into aliquots intended for RNAseq (five samples control; five samples IBL-302), mass spectrometry (see separate section), and Western blot verification. RNA was prepared using Ion AmpliSeq™ Transcriptome Human Gene Expression Kit Preparation protocol and sequenced on the Ion Proton™ System using the Ion PI™ Hi-Q Sequencing 200 Kit chemistry (200 bp read length, Thermo Fisher), as described in Braekeveldt *et al* (2018). Counts were normalized relative to total counts per sample, and the data were analyzed using the R2: Genomic Analysis and Visualization Platform (http://r2.amc.nl).

## Mass spectrometry

LU-NB-3 cells were seeded on laminin precoated plates (10 μg/ml laminin-521, Biolamina) and allowed to attach for 24 h. Cells were treated with 50 nM IBL-302 or vehicle control for 48 h before harvesting. Each sample consisted of pooled cells from plates treated equally, and samples were then divided into aliquots intended for RNAseq (see separate section), mass spectrometry (four samples control; three samples IBL-302), and Western blotting. A detailed description of the mass spectrometry analyses can be found in Appendix Supplementary Methods.

## Statistics

All values are reported as mean ± SEM from $n$ independent biological experiments, where $n$ is depicted in each figure. The two-sided Student's unpaired *t*-test was used for statistical analyses between two groups, while one-way ANOVA was used for statistical analyses between three groups or more. Three levels of significance were used: *$P < 0.05$; **$P < 0.01$; and ***$P < 0.001$. Publicly available

## The paper explained

### Problem

The phosphoinositol-3-kinase (PI3K) signaling pathway is constitutively active in many cancer types and can act as a major driver of the disease. Inhibition of PI3K and/or of its downstream effector mammalian target of rapamycin (mTOR) has shown promising results in multiple preclinical cancer models, including childhood cancer neuroblastoma. However, resistance via activation of parallel signaling pathways might occur. Furthermore, patients with high-risk neuroblastoma present with dismal prognosis, and neuroblastoma survivors often experience severe later life effects as a consequence of heavy chemotherapy at young age. Hence, novel anti-neuroblastoma therapies are urgently needed.

### Results

We developed multikinase PIM/PI3K/mTOR inhibitors, and among > 700 cancer cell lines of various origin, neuroblastoma was particularly sensitive toward the triple kinase inhibitor IBL-302. IBL-302 was more effective than single PI3K inhibitors, and multikinase inhibition induced apoptosis, neuronal differentiation, and downregulated N-Myc levels in neuroblastoma cell lines and patient-derived xenograft (PDX) cells. Treatment with IBL-302 *in vivo* reduced tumor growth in neuroblastoma xenografts. Furthermore, IBL-302 acted in synergy with the clinically used chemotherapeutic cisplatin to induce cell death and reduce tumor growth. Global genome and proteome analyses revealed that cell motility and apoptosis were major biological responses to multikinase targeting.

### Impact

Simultaneous targeting of PIM, PI3K, and mTOR is an effective strategy to induce differentiation, cell death, and reduced tumor growth in neuroblastoma. This multikinase targeting might help overcome PI3K treatment resistance. Furthermore, the addition of IBL-302 to current chemotherapeutic regimen could be a viable strategy to improve outcome and reduce severe late effects in neuroblastoma patients with high-risk tumors.

datasets containing 88 and 498 neuroblastomas, respectively (R2: microarray analysis and visualization platform; http://r2.amc.nl), were used to analyze gene expression and overall survival. Kaplan–Meier survival analyses for *in vivo* studies were generated using GraphPad Prism version 7.0c. Statistical significance was calculated using the log-rank test.

### Study approval

NMRI-Nu/Nu mice were housed in a controlled environment. The regional ethics committee approved all procedures (ethical permits M11-15 and M146-13).

## Data availability

The mass spectrometry proteomic data have been deposited to the ProteomeXchange Consortium via the PRIDE (Perez-Riverol *et al*, 2019) partner repository with the dataset identifier PXD014234 (https://www.ebi.ac.uk/pride/archive/projects/PXD014234).

The RNA sequencing data discussed in this publication have been deposited in NCBI's Gene Expression Omnibus (Edgar *et al*, 2002) and are accessible through GEO Series accession number GSE133137 (https://www.ncbi.nlm.nih.gov/geo/query/acc.cgi?acc=

GSE133137), and were analyzed using the R2: Genomics Analysis and Visualization Platform (http://r2.amc.nl).

**Expanded View** for this article is available online.

## Acknowledgements

This work was supported by funding from the Swedish Cancer Society (to SM, DB), the Swedish Research Council (to DB), the Swedish Childhood Cancer Fund (to SM, KvS, DB), Region Skåne and the research funds of Skåne University Hospital (to DB), the Mary Bevé Foundation (to SM, KvS, DB), Magnus Bergvalls stiftelse (to SM, DB), the Thelma Zoéga Foundation (to SM), Hans von Kantzow Foundation (to SM), Crafoord Foundation (to DB), Åke Wiberg Foundation (to DB), Jeanssons Stiftelser (to DB), Ollie och Elof Ericssons stiftelser (to DB), Berth von Kantzows stiftelse (to DB), the Royal Physiographic Society of Lund (to SM, DB), and the Spanish Ministry of Health and Social Policy (ADE08/90038) and the Spanish Ministry of Science and Innovation (CIT-090000-2008-14) (to JP, SMa, CBA). We would like to thank the Local MS Support at Medical Faculty, Lund University. The authors would like to acknowledge support of the National Genomics Infrastructure (NGI)/Uppsala Genome Center and UPPMAX for providing assistance in massive parallel sequencing and computational infrastructure. Work performed at NGI/Uppsala Genome Center has been funded by RFI/VR and Science for Life Laboratory, Sweden.

## Author contributions

SMo, KH, KR, MO, KS, and DB designed the research; SMo, KH, KR, CG-R, JE, and CW performed the research; MO contributed new reagents; JP designed the compounds IBL-202, IBL-301, and IBL-302; SMa and CB-A participated in the chemical synthesis/optimization of this series of inhibitors and their preclinical characterization; KS performed bioinformatic analyses; SMo, KH, KR, CW, KS, and DB analyzed data; SMo, KH, SMa, KS, and DB wrote the paper; and SMo and DB supervised the study. All authors read and approved the manuscript.

## Conflict of interest

Michael O'Neill is Director of Research and Development at Inflection Biosciences, who provided the IBL reagents.

## For more information

(i) https://www.cancerrxgene.org; *Genomics of Drug Sensitivity in Cancer*

(ii) https://hgserver1.amc.nl/cgi-bin/r2/main.cgi; *R2: Genomics Analysis and Visualization Platform*

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
