## [Review Process File · EMBO Molecular Medicine]

Anti-tumor effects of PIM/PI3K/mTOR triple kinase inhibitor IBL-302 in neuroblastoma

Sofie Mohlin, Karin Hansson, Katarzyna Radke, Sonia Martinez, Carmen Blanco-Apiricio, Cristian Garcia-Ruiz, Charlotte Welinder, Javanshir Esfandyari, Michael O'Neill, Joaquin Pastor, Kristoffer von Stedingk, and Daniel Bexell

Review timeline:

Submission to EMBO Journal:	30 October 2018
Editorial decision:	9 November 2018
Transfer to EMBO Mol Med:	12 November 2018
Editorial Decision:	6 December 2018
Revision received:	9 May 2019
Editorial Decision:	3 June 2019
Revision received:	4 June 2019
Accepted:	24 June 2019

Editor: Lise Roth

Transaction Report:

Decision from The EMBO Journal

9 November 2018

Thank you for submitting your manuscript (EMBOJ-2018-101036) to The EMBO Journal. I have now read your manuscript carefully and discussed it with my colleagues, and I regret to say that we cannot offer publication in The EMBO Journal. However, I encourage you to take advantage of our transfer option to our sister journal EMBO Molecular Medicine.

We appreciate your results presenting IBL-302 as first-in-class triple inhibitor co-targeting PI3K-mTOR-PIM-dependent progression of neuroblastoma. We appreciate that these findings will be of interest to the field. However, we also noted that PIM kinase overexpression and inhibition have been reported to confer resistance to PI3K inhibition. Also, synergism between PIM and PI3K inhibition has been demonstrated. In addition, the underlying details of how the novel inhibitor acts at molecular and cellular level remain rather unclear in our view. Thus, despite its notable aspects, your manuscript does in our view not provide the sufficiently striking conceptual advance and conclusive mechanistic insights that we have to ask for publication in The EMBO Journal and I am therefore sorry to say that we have decided not to send it out for peer-review.

Following your transfer statement, I have enquired and discussed back with my colleague at EMBO Molecular Medicine, Lise Roth, and thus encourage you to make use of our transfer option, in case of which the work would be considered by external peer-review..

I regret to have to disappoint you this time and I hope that you will take advantage of our transfer option.

Decision from EMBO Molecular Medicine

6 December 2018

Thank you for the submission of your manuscript to EMBO Molecular Medicine. We have now heard back from the three referees who were asked to evaluate your manuscript.

As you will see from the reports below, while they all mention the interest and potential clinical relevance of the study, the referees also raise substantial concerns on your work, which should be convincingly addressed in a major revision of the present manuscript. In particular, the text should be revised (overstatements throughout the manuscript, need for improving the introduction (missing references) and the discussion), and additional experiments should be performed: requirement for additional cell lines in vitro, effects of IBL302 in vitro, in-depth apoptosis analyses and mechanistic studies in general, missing controls, additional in vivo models.

Addressing the reviewers' concerns in full will be necessary for further considering the manuscript in our journal, and acceptance of the manuscript will entail a second round of review. EMBO Molecular Medicine encourages a single round of revision only and therefore, acceptance or rejection of the manuscript will depend on the completeness of your responses included in the next, final version of the manuscript. For this reason, and to save you from any frustrations in the end, I would strongly advise against returning an incomplete revision.

EMBO Molecular Medicine has a "scooping protection" policy, whereby similar findings that are published by others during review or revision are not a criterion for rejection. Should you decide to submit a revised version, I do ask that you get in touch after three months if you have not completed it, to update us on the status. Please also contact us as soon as possible if similar work is published elsewhere. If other work is published, we may not be able to extend the revision period beyond three months.

I look forward to receiving your revised manuscript.

***** Reviewer's comments *****

Referee #1 (Remarks for Author):

The paper by S. Mohlin et al reports the anti-tumor activity of a triple kinase inhibitor (IBL-302) targeting PIM/PI3K/mTOR in neuroblastoma.

The authors used a new compound targeting PIM/PI3K/mTOR to screen a large panel of cancer cell lines. They identified neuroblastoma, a pediatric cancer of the peripheral nervous system as showing the highest sensitivity to IBL-302. They compared its activity to inhibitors targeting only one specific kinase and showed that IBL-302 induces apoptosis, differentiation and MYCN protein decrease. Then, they determined the effect of combined treatment with IBL-302 and standard chemotherapy. RNA-seq and proteomic analysis of treated samples were performed followed by global analysis indicating modifications of various biological processing upon treatment.

Major comments

1. The authors claim that "Neuroblastoma was by far the most sensitive cancer to IBL-302 treatment (Fig. 2A)". This is not so clear from the graph shown on figure 2A, but the y-axis presents the Log₁₀ GI50 of IBL-302, and it would be easier for the reader to see the GI50 values directly compared according to the samples. Also, Figure 2B used LN mM GI50 which is a different unit and does not facilitate comparison with Figure 2A.
2. The effect of IBL-202 and IBL-301 on pAkt and pS6K is shown only for LU-NB-3 cells, that are derived from a neuroblastoma PDX. It is not clear why this sample has been selected for this analysis. Many neuroblastoma cell lines have been described in the literature and extensively characterized; at least several of them should be included in this analysis. The authors have such cell lines in hands as they are used in Figure 2B. The same remark is true for the analysis of cell cycle, apoptosis, differentiation and measure of MYCN levels.
3. The authors show a WB of cleaved-caspase to evaluate apoptosis, again in only one cell line. It is not possible to evaluate the percent of cells undergoing apoptosis by such analysis; FACS analysis using annexin V for example should be used to get a more precise evaluation of apoptosis. Also, the impact of IBL-202 and 301 on differentiation should be evaluated more precisely and quantified.
4. The authors write : "Due to superior drug properties over IBL-301, IBL-302, a compound with comparable structure but increased bioavailability, was chosen for further testing." There are no data showing results leading to this conclusion.

5. The effect of IBL-302 on pAkt and pS6K should be shown for the 3 analyzed cell lines and additional neuroblastoma cell lines currently studied as mentioned in point 2.
6. Low doses of cisplatin and IBL-302 were then evaluated in combination in vitro and in vivo on LU-NB-3 PDX cells. Again only one neuroblastoma sample has been studied. Five mice were treated for the in vivo experiment, among which 3 did not respond to the combination and 2 responded, which renders the interpretation difficult and interrogates on the efficacy of the combination.
7. Figure 6 is of bad quality and it is not possible to read the gene names and GO categories.
8. PIM3 is not evaluated at the expression level and in terms of its kinase activity.

Other comments

8. The discussion is poor.
9. The authors claim that PIM3 high expression is associated with adverse patient outcomes in neuroblastoma. However, they show that PIM3 expression is higher in MYCN amplified tumors. Since MYCN amplification is a very strong prognosis marker in neuroblastoma, univariate analysis on PIM3 may reflect in fact MYCN status. Multivariate analysis would be required to search for a potential effect of PIM3 expression on outcome, independently of MYCN status. If PIM3 is important in neuroblastoma oncogenesis with respect to its kinase activity and involvement in the PI3K/mTOR pathway, expression at the mRNA levels is not necessarily a relevant surrogate marker.
10. The authors should be more precise in the introduction regarding previous knowledge on targeting the PI3K/mTOR in neuroblastoma. Only one sentence referring to 9 different papers is mentioned.

Referee #2 (Remarks for Author):

The authors have evaluated the efficacy of IBL-302, an inhibitor of the PIM, PI3K, and mTOR kinases, in neuroblastoma cell lines and xenograft tumors. The manuscript presents some interesting data and suggests the potential efficacy of IBL-302 in neuroblastoma. There are several key limitations and other issues that need to be addressed:

- The authors have not included several important references exploring the role of PI3K/mTOR in neuroblastoma, including multiple references about the efficacy of perifosine and SF1126, among many others
- The PIM expression data presented (Fig 1A, B) are relatively meaningless without comparison to control cells/tissues
- the Kaplan-Meier curves presented (Fig 1C) do not indicate a particularly strong effect of PIM expression on patient outcomes
- the induction of apoptosis (Fig 3E-G) does not appear to be robust (particularly the Caspase cleavage assay). Alternative responses to IBL-302 should be considered
- The clinical/translational relevance of these findings are somewhat unclear - what are the drug levels achieved in the mouse xenograft models? And are these drug levels (IC50 values determined in the manuscript) potentially achievable with this agent in human patients?
- the use of subcutaneous tumor models limits the significance of the findings somewhat; orthotopic or PDX models would be more relevant
- the in vivo responses to IBL-302 (Fig 4E-F) do not appear to be durable, raising a question about the authors' hypothesis that IBL-302 is able to overcome PI3K resistance. Comparisons against other single PI3K inhibitors or other studies to demonstrate PI3K resistance in the animal models are needed

Referee #3 (Comments on Novelty/Model System for Author):

Neuroblastoma is a difficult tumor to treat. While chemotherapy is somewhat effective in this disease, it has a lot of toxicity and the tumors often recur. Finding ways to improve the efficacy of current treatments by targeting important pathways would have a positive impact on this disease. The *in vitro* and *in vivo* models the authors use are appropriate. It is good they are using some patient-derived tumor cells as well as multiple cell lines to make sure what they are seeing isn't particular to a single cell line. They also verify their findings in a number of different ways to support their conclusions.

Referee #3 (Remarks for Author):

In this article, the authors examine the impact of the triple kinase inhibitor, IBL-302, on the treatment of neuroblastoma. These kinase inhibitors were identified through a drug sensitivity screen of 47 different tumor types and it was found that neuroblastoma was the most sensitive tumor to treatment with IBL-302. Interestingly, the neuroblastoma cell line models the group used were not particularly sensitive to the individual tyrosine kinase inhibitors but showed considerable sensitivity to the triple-kinase inhibitor. They demonstrate that neuroblastoma expresses increased levels of PIM3 and that high expression of PIM3 in neuroblastoma correlates with a worse prognosis suggesting an important role of this pathway. The authors demonstrate inhibition of Akt and PRAS40 phosphorylation supporting the assertion that IBL-302 (and related compounds) are capable of inhibiting these pathways. They also demonstrate induction of apoptosis and differentiation in response to treatment with their kinase inhibitors. Additionally, these compounds appear to decrease N-Myc protein levels in the cells. Given the prominent role of N-Myc in neuroblastoma, this could be very beneficial. In an *in vivo* mouse model, IBL-302 is shown to inhibit tumor growth and prolong survival suggesting activity in neuroblastoma. More importantly, combining IBL-302 with commonly-used chemotherapy agents improved the effects of both drugs even leading to growth suppression. The suggestion from the authors is that patients with neuroblastoma could obtain similar benefits while using less chemotherapy when combined with IBL-302. Given the high toxicity associated with neuroblastoma treatment, this would be a welcome advance.

Overall, the article is clear and well written. The authors outline their rationale for pursuing these studies and it is well laid out and easy to follow. There is a lot of detail (perhaps even a bit too much detail) in the descriptions of the experiments and figures. The experiments have appropriate controls and use multiple cell lines including patient-derived cells, which tend to be a better model for *in vitro* and *in vivo* experiments. While the interpretation of the data is appropriate, I would recommend avoiding use of superlative words like "by far" and "profound". While I agree that these kinase inhibitors are showing a beneficial effect (and demonstrating statistical significance), many of the effects are rather modest and it's best not to overstate. Some minor comments:

1. While it was explained in the article why IBL-302 was used for *in vivo* testing and there are some data provided replicating the *in vitro* studies, it isn't clear why IBL-302 wasn't used for the detailed analysis of the kinase inhibition and its impact (figure 3). Is it not as workable in the *in vitro* setting vs IBL-202 and IBL-301?
2. The authors tested individual kinase inhibitors against their neuroblastoma tumor cells and found minimal effect. It would be interesting to see if combining targeted kinase inhibitors against PIM, PI3K, and mTOR could replicate the findings of their triple-kinase inhibitors. That would lend more support to the conclusion inhibition of multiple kinases is required to have a positive effect on neuroblastoma treatment (rather than there being an off-target effect particular to IBL-302).
3. The improved effect of combining IBL-302 with conventional chemotherapy is encouraging, but the impact on IC50 is pretty modest. It would be interesting to perform isobolographic analysis to ascertain whether these compounds are having a synergistic or additive effect.
4. Figure 6 is very difficult to read given the small typeface. It may be helpful to break up this figure a bit to enable some of the text to be larger.

Referee #1 (Remarks for Author):

The paper by S. Mohlin et al reports the anti-tumor activity of a triple kinase inhibitor (IBL-302) targeting PIM/PI3K/mTOR in neuroblastoma. The authors used a new compound targeting PIM/PI3K/mTOR to screen a large panel of cancer cell lines. They identified neuroblastoma, a pediatric cancer of the peripheral nervous system as showing the highest sensitivity to IBL-302. They compared its activity to inhibitors targeting only one specific kinase and showed that IBL-302 induces apoptosis, differentiation and MYCN protein decrease. Then, they determined the effect of combined treatment with IBL-302 and standard chemotherapy. RNA-seq and proteomic analysis of treated samples were performed followed by global analysis indicating modifications of various biological processing upon treatment.

Major comments

1. The authors claim that "Neuroblastoma was by far the most sensitive cancer to IBL-302 treatment (Fig. 2A)". This is not so clear from the graph shown on figure 2A, but the y-axis presents the Log10 GI50 of IBL-302, and it would be easier for the reader to see the GI50 which is a different unit and does not facilitate comparison with Figure 2A.

We thank the reviewer for noting these dissimilarities. We have accordingly changed the y-axis units in both figures to GI50 values (**new Figure 2A-B**).

2. The effect of IBL-202 and IBL-301 on pAkt and pS6K is shown only for LU-NB-3 cells that are derived from a neuroblastoma PDX. It is not clear why this sample has been selected for this analysis. Many neuroblastoma cell lines have been described in the literature and extensively characterized; at least several of them should be included in this analysis. The authors have such cell lines in hands as they are used in Figure 2B. The same remark is true for the analysis of cell cycle, apoptosis, differentiation and measure of MYCN levels.

We understand the reviewer's concern that these results might be cell line-specific and have according to suggestions added data from one other PDX (LU-NB-2) as well as two conventional neuroblastoma cell lines, SK-N-BE(2)c and SK-N-SH. We've added data for effects on pAkt, cell death, differentiation and N-Myc western blot (**new Figures 3A-B, 3F-G, 4A, 4E-F, and Expanded View Figure 1B-E**). There are several previous reports that support the notion of reduced N-Myc levels following PI3K/mTOR inhibition in neuroblastoma (e.g. Cage et al., 2015; Chanthery et al., 2012; Chesler et al., 2006; Erdreich- Epstein et al., 2017; Johnsen et al., 2008; Vaughan et al., 2016). Overall, the addition of these new data shows that our findings are general and not cell line-specific.

3. The authors show a WB of cleaved-caspase to evaluate apoptosis, again in only one cell line. It is not possible to evaluate the percent of cells undergoing apoptosis by such analysis; FACS analysis using annexin V for example should be used to get a more precise evaluation of apoptosis. Also, the impact of IBL-202 and 301 on differentiation should be evaluated more precisely and quantified.

We appreciate the encouragement to more precisely demonstrate the apoptotic and differentiation-inducing effects from these inhibitors. We have accordingly performed FACS analysis using Annexin V for direct quantification of cell death (**new Figure 4E-F**). We also provide morphological evidence of differentiation from additional cell lines and have further quantified neurite outgrowth as a measurement of neuronal differentiation (**new Figure 3D-E and Expanded View Figure 2D-E**).

4. The authors write: "Due to superior drug properties over IBL-301, IBL-302, a compound with comparable structure but increased bioavailability, was chosen for further testing." There are no data showing results leading to this conclusion.

The information on in vivo properties of IBL-301 and IBL-302 is confidential from Inflection Biosciences Ltd and we can hence not elaborate on this further. However, to avoid mentioning something that we cannot provide details for, we have accordingly changed the text to: "IBL-302, a compound with comparable structure to IBL-301 but with increased bioavailability was chosen for further testing."

5. The effect of IBL-302 on pAkt and pS6K should be shown for the 3 analyzed cell lines and additional neuroblastoma cell lines currently studied as mentioned in point 2.

Effects of IBL-302 on pAkt (s473 and t308) levels are analyzed in several PDX- and conventional

neuroblastoma cell lines and again show that our findings are general and not cell line-specific (**new Figure 5A-B** and **Appendix Figure S2A-B**). We have tried to repeat the pS6K western blots in several cell lines but due to technical issues with the antibodies we were not able to include more data regarding this.

6. Low doses of cisplatin and IBL-302 were then evaluated in combination in vitro and in vivo on LU-NB-3 PDX cells. Again only one neuroblastoma sample has been studied. Five mice were treated for the in vivo experiment, among which 3 did not respond to the combination and 2 responded, which renders the interpretation difficult and interrogates on the efficacy of the combination.

To strengthen our conclusion that there is a benefit to combine cisplatin and IBL-302 we have now treated additional cell lines (SK-N-BE(2)c and SK-N-SH) with the combination of these drugs (**new Figure 6B**). We have further performed isobolographic analyses (as suggested by Reviewer #3) and show that there is a synergistic effect from the combination drugs (**new Figure 6B**).

7. Figure 6 is of bad quality and it is not possible to read the gene names and GO categories. We appreciate that the reviewer highlights this issue. We have accordingly improved the quality of the figure so that gene names and GO categories are readable (**new Figure 7**).

8. PIM3 is not evaluated at the expression level and in terms of its kinase activity.

We have repeatedly tried to detect PIM3 as well as downstream pBad levels on western blot but have not been able to do so due to technical issues with the antibodies. We do observe a modest correlation between PIM3 and outcome, and that PIM3 seems to be expressed at a higher level than PIM1 and PIM2 in neuroblastoma (**old Figures 1A-B** and **Appendix Figure S1A-B**). However, since we are not able to determine the activity of PIM3 in our neuroblastoma cells, we cannot draw any certain conclusions on PIM3 specifically and have now removed **old Figures 1A-B** from the manuscript, and transferred **old Figure 1C** to **new Appendix Figure S1C-D**.

Other comments

8. The discussion is poor.

We have attempted to address this by removing text and being clearer as well as soften our statements (as suggested by Reviewer #3).

9. The authors claim that PIM3 high expression is associated with adverse patient outcomes in neuroblastoma. However, they show that PIM3 expression is higher in MYCN amplified tumors. Since MYCN amplification is a very strong prognosis marker in neuroblastoma, univariate analysis on PIM3 may reflect in fact MYCN status. Multivariate analysis would be required to search for a potential effect of PIM3 expression on outcome, independently of MYCN status. If PIM3 is important in neuroblastoma oncogenesis with respect to its levels is not necessarily a relevant surrogate marker.

We agree that mRNA expression levels are not necessarily a relevant marker for possible correlations to outcome etc. We have, despite repeated attempts, been able to detect pBad (downstream surrogate marker of PIM activity) levels using western blot due to technical difficulties with the antibodies, and have moved data on PIM mRNA expression and outcome to **new Appendix Figure S1C-D**. To investigate if the potential effects that PIM expression might still have on outcome, we performed multivariate analysis correcting for MYCN status, and PIM3 did indeed fall out as a significant independent prognostic variable while PIM1 did not. These analyses have been added to the text and **new Expanded View Figure 2C**.

10. The authors should be more precise in the introduction regarding previous knowledge on targeting the PI3K/mTOR in neuroblastoma. Only one sentence referring to 9 different papers is mentioned.

We have added text and references describing the effects of targeting PI3K/mTOR in neuroblastoma more thoroughly. To better present the effects of PI3K/mTOR inhibition, we have specifically divided the references and described their findings.

Referee #2 (Remarks for Author):

The authors have evaluated the efficacy of IBL-302, an inhibitor of the PIM, PI3K, and mTOR kinases, in neuroblastoma cell lines and xenograft tumors. The manuscript presents some interesting data and suggests the potential efficacy of IBL-302 in neuroblastoma.

We thank the reviewer for the interest in our paper.

There are several key limitations and other issues that need to be addressed:

- The authors have not included several important references exploring the role of PI3K/mTOR in neuroblastoma, including multiple references about the efficacy of perifosine and SF1126, among many others

We have added text and references on PI3K/mTOR targeting in neuroblastoma from a variety of PI3K inhibitors in the introduction.

- The PIM expression data presented (Fig 1A, B) are relatively meaningless without comparison to control cells/tissues

- the Kaplan-Meier curves presented (Fig 1C) do not indicate a particularly strong effect of PIM expression on patient outcomes

We agree with the reviewer that data on PIM expression as well as correlations to outcome do not add value without being compared to normal tissue or determining the activity of the PIM proteins (a point raised by Reviewer #1). We are unable to compare PIM expression to normal tissues and can neither detect PIM proteins or downstream target pBad on western blot due to technical issues with the antibody and have consequently removed the data on expression levels of PIM and PI3K family members (**old Figures 1AB**) from the manuscript. We have however kept the Kaplan-Meier curves showing the correlations between PIM1 and PIM3 and outcome. Although these data are only correlative, they fall out significant for PIM3 and we believe that it is worth presenting.

These data are moved to **new Appendix Figure S1C-D** to put less focus on the results. To determine whether PIM1 and PIM3 predict outcome also independent of MYCN status, we performed a multivariate analysis and while PIM3 indeed fall out significant, PIM1 does not (**new Expanded View Figure 2C**).

- the induction of apoptosis (Fig 3E-G) does not appear to be robust (particularly the Caspase cleavage assay). Alternative responses to IBL-302 should be considered

We thank the reviewer for highlighting this issue. We have analyzed cell death using the additional Annexin V method in multiple neuroblastoma cell lines (**new Figure 4E-F** (IBL-202 and IBL-301); **new Figure 5G** (IBL-302)).

- The clinical/translational relevance of these findings are somewhat unclear - what are the drug levels achieved the mouse xenograft models? And are these drug levels (IC50 values determined in the manuscript) potentially achievable with this agent in human patients?

We agree with the reviewer that such translations would be extremely valuable, but such data cannot be retrieved. The doses given to mice are determined by ethical concerns where mice are allowed to lose only 10% of their body weight. In the clinic, the doses possible to use for patients are determined in Phase I clinical studies.

- the use of subcutaneous tumor models limits the significance of the findings somewhat; orthotopic or PDX models would be more relevant

We have indeed performed the cisplatin/IBL-302 in vivo study (**new Figure 6E-H**) using our PDX model. We do however agree with the reviewer that there are limitations to using subcutaneous mouse models but have chosen to do so to allow for accessible continuous measurement of tumor growth.

- the in vivo responses to IBL-302 (Fig 4E-F) do not appear to be durable, raising a question about the authors' hypothesis that IBL-302 is able to overcome PI3K resistance. Comparisons against other single PI3K inhibitors or other studies to demonstrate PI3K resistance in the animal models are needed

We appreciate that the reviewer raises a discussion on the matter of putative advantages of using triple kinase inhibitors over single PI3K inhibitors, a point also raised by reviewer #3 (see **new Expanded View Figure 3A-E** where we compare single PIM, PI3K or mTOR inhibitors with the combination of all three inhibitors). This is a difficult matter to elucidate since triple kinase

inhibitors such as IBL-302 might present additional advantages than just inhibiting each target separately. We have approached this issue by treating several cell lines (SK-N-FI and SK-N-AS) with IBL-302 as well as PI3K inhibitors PI-103 and Dactolisib in vitro and can show that the effects from IBL-302 is greater than that of single PI3K inhibitors. When translating this into in vivo experiments there are no significant differences between the treatments. There are several plausible explanations including dosing issues and choice of PI3K single target inhibitor. These data have been added to the manuscript (**new Expanded View Figure 3F-G**) and is discussed in the text.

Referee #3 (Comments on Novelty/Model System for Author):

Neuroblastoma is a difficult tumor to treat. While chemotherapy is somewhat effective in this disease, it has a lot of toxicity and the tumors often recur. Finding ways to improve the efficacy of current treatments by targeting important pathways would have a positive impact on this disease. The in vitro and in vivo models the authors use are appropriate. It is good they are using some patient-derived tumor cells as well as multiple cell lines to make sure what they are seeing isn't particular to a single cell line. They also verify their findings in a number of different ways to support their conclusions.

We thank the reviewer for these encouraging comments.

Referee #3 (Remarks for Author):

In this article, the authors examine the impact of the triple kinase inhibitor, IBL-302, on the treatment of neuroblastoma. These kinase inhibitors were identified through a drug sensitivity screen of 47 different tumor types and it was found that neuroblastoma was the most sensitive tumor to treatment with IBL-302. Interestingly, the neuroblastoma cell line models the group used were not particularly sensitive to the individual tyrosine kinase inhibitors but showed considerable sensitivity to the triple-kinase inhibitor. They demonstrate that neuroblastoma expresses increased levels of PIM3 and that high expression of PIM3 in neuroblastoma correlates with a worse prognosis suggesting an important role of this pathway. The authors demonstrate inhibition of Akt and PRAS40 phosphorylation supporting the assertion that INL-302 (and related compounds) are capable of inhibiting these pathways. They also demonstrate induction of apoptosis and differentiation in response to treatment with their kinase inhibitors. Additionally, these compounds appear to decrease N-Myc protein levels in the cells. Given the prominent role of N-Myc in neuroblastoma, this could be very beneficial. In an in vivo mouse model, IBL-302 is shown to inhibit tumor growth and prolong survival suggesting activity in neuroblastoma. More importantly, combining IBL-302 with commonly-used chemotherapy agents improved the effects of both drugs even leading to growth suppression. The suggestion from the authors is that patients with neuroblastoma could obtain similar benefits while using less chemotherapy when combined with IBL-302.

Given the high toxicity associated with neuroblastoma treatment, this would be a welcome advance. Overall, the article is clear and well written. The authors outline their rationale for pursuing these studies and it is well laid out and easy to follow. There is a lot of detail (perhaps even a bit too much detail) in the descriptions of the experiments and figures. The experiments have appropriate controls and use multiple cell lines including patient-derived cells, which tend to be a better model for in vitro and in vivo experiments. While the interpretation of the data is appropriate, I would recommend avoiding use of superlative words like "by far" and "profound". While I agree that these kinase inhibitors are showing a beneficial effect (and demonstrating statistical significance), many of the effects are rather modest and it's best not to overstate.

We agree with the reviewer that overstatements are unnecessary. We have accordingly softened our statements throughout the text.

Some minor comments:

1. While it was explained in the article why IBL-302 was used for in vivo testing and there are some data provided replicating the in vitro studies, it isn't clear why IBL-302 wasn't used for the detailed analysis of the kinase inhibition and its impact (figure 3). Is it not as workable in the in vitro setting vs IBL-202 and IBL-301?

We understand the reviewer's concern on this issue. IBL-202 and IBL-301 were simply generated first line, and we have hence used these for extensive in vitro characterization. Due to suboptimal effects of IBL-301 in vivo (unsatisfying uptake), IBL-302 was synthesized. IBL-302 is a close analogue to IBL-301. IBL-302 is workable in vitro (**old Figures 4A-F and Appendix Figure S2B**).

We have expanded the *in vitro* data on IBL-302 for a detailed analysis on downstream target effects in two cell lines (LU-NB-3 and SK-N-BE(2)c, **new Figure 5A-B**), morphological and quantitative data on differentiation in three cell lines (LU-NB-3, SK-N-BE(2)c and SK-N-SH; **new Figure 5C-D and Appendix Figure S2C**), N-Myc protein downregulation in two cell lines (LU-NB-3 and LU-NB-2, **new Figure 5E**), and CellTiterGlo as well as Annexin V analyses in four cell lines (LU-NB-2, LU-NB-3, SK-N-BE(2)c and SK-N-SH; **new Figure 5F-G**). Thus, IBL-301 and IBL-302 display similar effects *in vitro* and we appreciate that we got the chance to expand and clarify this further.

2. The authors tested individual kinase inhibitors against their neuroblastoma tumor cells and found minimal effect. It would be interesting to see if combining targeted kinase inhibitors against PIM, PI3K, and mTOR could replicate the findings of their triple-kinase inhibitors. That would lend more support to the conclusion inhibition of multiple kinases is required to have a positive effect on neuroblastoma treatment (rather than there being an off-target effect particular to IBL-302).

We appreciate that the reviewer raises this discussion point. One concern with this kind of experiment is that combinatorial treatment with three individual drugs is not completely comparable to treatment with a single triple target inhibitor, as the latter might have additional beneficial effects. However, we have added data where we treat cells with PI3K only, PIM only, mTOR only or the combination of these inhibitors and analyzed the outcome using the Annexin V assay. We do observe a greater effect when combining these inhibitors as compared to any individual inhibitor. However, the effects are modest, putatively explained by additional beneficial effects than just targeting each protein individually and/or dosing issues. Studies determining whether there are beneficial effects would indeed be interesting but is beyond the scope of this paper. These data are presented in **new Expanded View Figure 5A-E**.

3. The improved effect of combining IBL-302 with conventional chemotherapy is encouraging, but the impact on IC50 is pretty modest. It would be interesting to perform isobolographic analysis to ascertain whether these compounds are having a synergistic or additive effect.

We thank the reviewer for raising this very interesting point. We have performed isobolographic analyses and in the case of Doxorubicin and Etoposide, the observed effects are additive (**new Appendix Figure S3A-B**). However, when cisplatin is combined with IBL-202, IBL-301, or IBL-302 there is synergistic effects (**new Figure 6A-B**).

4. Figure 6 is very difficult to read given the small typeface. It may be helpful to break up this figure a bit to enable some of the text to be larger.

We thank the reviewer for noticing this issue. We have accordingly improved the readability as well as quality of the figure (**new Figure 7**).

2nd Editorial Decision

3 June 2019

Thank you for the submission of your revised manuscript to EMBO Molecular Medicine. We have now heard back from the three reviewers who were asked to reassess your manuscript. As you will see from the enclosed reports, all the referees are appreciative of the considerable improvements in the manuscript, however referee #1 still has questions that should be addressed in a minor revision of the present manuscript (please also address the comments from referee #3). At this stage, we'd like you to discuss the referees' points in writing, and we do not ask you to provide any additional experiments. Please provide a letter including my comments, the reviewers' reports and your detailed responses to their comments.

I look forward to reading a new revised version of your manuscript as soon as possible.

***** Reviewer's comments *****

Referee #1 (Remarks for Author):

The authors have answered a number of remarks in a convincing way. Yet, some questions remain without answers for various reasons: some information on the IBL-302 compound are confidential, WB for pS6K have not been added due to technical issues, the kinase activity of PIM3 could not be evaluated. Regarding answer to point 8, the authors show some qPCR

data for the expression of PIM1 and PIM3 (Appendix Figure S1A-B) and write in their answer that PIM 3 seems to be expressed at higher level than PIM1 and PIM2 in neuroblastoma. Since qPCR uses different primers and probes for each gene, it is not possible to compare the expression of the different genes, this is not correct. Using this technique, one can compare different samples using the same probes with respect to a common reference, but different genes cannot be compared.

Only one PDX model has been used to evaluate *in vivo* the combination between IBL-302 and cisplatin, with 2 mice showing a good response and 3 mice having not responded. This effect is rather mild to provide a strong pre-clinical rationale for further treatment of neuroblastoma patients. Figure 7 is now of better quality but the message provided by the analysis of the global transcriptome, proteome and phospho-proteome on the effect of IBL-302 treatment does not provide much interesting information. The identification of apoptosis, programmed cell death and cell cycle does not bring much with respect to the effect of the compound that induces apoptosis and differentiation.

The discussion is still poor, with very general statements and no in-depth discussion of specific points with regards to precise data of the literature.

Referee #2 (Remarks for Author):

the authors have adequately addressed all of my comments and concerns from the initial submission

Referee #3 (Comments on Novelty/Model System for Author):

I felt the models used were appropriate. I appreciated the expansion of cell lines tested in the revised version of the article.

Referee #3 (Remarks for Author):

Comments for authors

In this revised article, the authors examine the impact of the triple kinase inhibitor, IBL-302, on the treatment of neuroblastoma. These kinase inhibitors were identified through a drug sensitivity screen of 47 different tumor types and it was found that neuroblastoma was the most sensitive tumor to treatment with IBL-302. Interestingly, the neuroblastoma cell line models the group used were not particularly sensitive to the individual tyrosine kinase inhibitors but showed considerable sensitivity to the triple-kinase inhibitor. They demonstrate that neuroblastoma expresses increased levels of PIM3 and that high expression of PIM3 in neuroblastoma correlates with a worse prognosis suggesting an important role of this pathway. Given the correlation of PIM3 expression with MYCN amplification, the authors performed multivariate analysis and interestingly demonstrate an independent prognostic effect of PIM3 on neuroblastoma outcomes. The authors demonstrate inhibition of Akt phosphorylation in an expanded panel of cell lines supporting the assertion that IBL-302 (and related compounds) are capable of inhibiting these pathways. Unfortunately, due to technical difficulties, they were not able to show direct effects on PIM or downstream targets. They also more definitively demonstrate induction of apoptosis and differentiation in response to treatment with their kinase inhibitors. Additionally, these compounds appear to decrease N-Myc protein levels in the cells. Given the prominent role of N-Myc in neuroblastoma, this could be very beneficial. In an *in vivo* mouse model, IBL-302 is shown to inhibit tumor growth and prolong survival suggesting activity in neuroblastoma. More importantly, combining IBL-302 with commonly-used chemotherapy agents improved the effects of both drugs even leading to growth suppression. These conclusions are strengthened by the accompanying isobolographic analysis. The suggestion from the authors is that patients with neuroblastoma could obtain similar benefits while using less chemotherapy when combined with IBL-302. Given the high toxicity associated with neuroblastoma treatment, this would be a welcome advance.

Overall, the article is clear and well written. Once again, the authors outline their rationale for pursuing these studies is well laid out and easy to follow. I appreciate that the authors have taken the time to address the comments from the reviewers and augmented their data as requested. They have increased their references in regard to PI3K/mTOR pathway in neuroblastoma. The addition of more cell lines in the response assays helps demonstrate that they are not seeing cell-line specific effects. Unfortunately, the direct effect of IBL-302 on PIM and the downstream targets could not be

measured directly, but not all proteins are amenable to Western analysis. I also liked the added experiment combining the targeted inhibitors (figure EV3), which showed some improved effect of the combination but not to the level seen with IBL-302. Also, by adding the Annexin V data, the apoptosis effects are more convincing. I appreciated the addition of the isobolographic analysis, which supports their conclusions regarding combination chemotherapy. Finally, they have softened the language (especially in the discussion) so it now aligns nicely with the data presented. In total, I feel the changes to the revised version strengthened the paper.

Minor Comment

1. In figure 6B, the addition of IBL-302 to low-dose cisplatin appears to have a synergistic effect in the LU-NB-3 and SK-N-BE(2)c cells. However, the SK-N-SH cells have a much different pattern of response and appear to possibly have an antagonistic effect. Do the authors have a possible cause for the different pattern seen in this cell line?

2nd Revision - authors' response

4 June 2019

Referee #1 (Remarks for Author):

The authors have answered a number of remarks in a convincing way.

We thank the reviewer for this positive response to our revision.

Yet, some questions remain without answers for various reasons: some information on the IBL-302 compound are confidential, WB for pS6K have not been added due to technical issues, the kinase activity of PIM3 could not be evaluated. Regarding answer to point 8, the authors show some qPCR data for the expression of PIM1 and PIM3 (Appendix Figure S1A-B) and write in their answer that PIM 3 seems to be expressed at higher level than PIM1 and PIM2 in neuroblastoma. Since qPCR uses different primers and probes for each gene, it is not possible to compare the expression of the different genes, this is not correct. Using this technique, one can compare different samples using the same probes with respect to a common reference, but different genes cannot be compared.

We thank the reviewer for this discussion and agree that qPCR cannot be used to compare different genes. Our answer was referring to data included in the original manuscript based on data derived from the R2 database. We do not state in the revised manuscript that PIM3 is expressed at higher level since these data were removed from the original submission.

Only one PDX model has been used to evaluate in vivo the combination between IBL-302 and cisplatin, with 2 mice showing a good response and 3 mice having not responded. This effect is rather mild to provide a strong pre-clinical rationale for further treatment of neuroblastoma patients. We do not overstate these results in the text but rather present the results as they stand, with two subgroups responding to treatment differently.

We think our results suggest that IBL-302 in combination with conventional chemotherapy has potential as treatment strategy for high-risk neuroblastoma. The translation of preclinical findings to clinical testing is of course very difficult to assess and includes various aspects involving preclinical models and drug dosing.

Here, we observed effects in multiple preclinical neuroblastoma models, which strengthen our findings. It is also important to note that we used significantly lower drug concentrations for both IBL-302 and Cisplatin in the PDX in vivo study. Thus, it might be possible to increase drug concentrations and obtain even stronger effects in vivo. We are however aware of the diverse response in the PDX in vivo study and have carefully gone through the terminology in the results and discussion sections not to make overstatements.

Figure 7 is now of better quality but the message provided by the analysis of the global transcriptome, proteome and phospho-proteome on the effect of IBL-302 treatment does not provide much interesting information. The identification of apoptosis, programmed cell death and cell cycle does not bring much with respect to the effect of the compound that induces apoptosis and differentiation.

We regret that the reviewer does not find these analyses informative. However, we believe they constitute an important part of the manuscript showing global effects on our cells and enriching for pathways that are in conformity with our experimental data. In addition, providing downstream

affected genes and pathways opens up for further analyses, by us and/or others, to identify mediators and important proteins involved in these mechanisms. Thus, we do agree that it would be interesting to further investigate the mechanisms of PIM/PI3K inhibition in neuroblastoma.

The discussion is still poor, with very general statements and no in-depth discussion of specific points with regards to precise data of the literature.

We regret that the reviewer does not find our discussion well written. In contrast, referee #3 writes *“Overall, the article is clear and well written ... Finally, they have softened the language (especially in the discussion) so it now aligns nicely with the data presented”*. Nevertheless, we now discuss, with added references, other studies using PIM inhibitors to target MYC-associated cancer types as well as combined PIM/PI3K inhibition strategies in cancer.

Referee #2 (Remarks for Author):

the authors have adequately addressed all of my comments and concerns from the initial submission
We thank referee #2 for this very concise and positive evaluation.

Referee #3 (Comments on Novelty/Model System for Author):

I felt the models used were appropriate. I appreciated the expansion of cell lines tested in the revised version of the article.

We thank referee #3 for these positive comments.

Referee #3 (Remarks for Author):

Comments for authors

In this revised article, the authors examine the impact of the triple kinase inhibitor, IBL-302, on the treatment of neuroblastoma. These kinase inhibitors were identified through a drug sensitivity screen of 47 different tumor types and it was found that neuroblastoma was the most sensitive tumor to treatment with IBL-302.

Interestingly, the neuroblastoma cell line models the group used were not particularly sensitive to the individual tyrosine kinase inhibitors but showed considerable sensitivity to the triple-kinase inhibitor. They demonstrate that PIM3 in neuroblastoma correlates with a worse prognosis suggesting an important role of this pathway. Given the correlation of PIM3 expression with MYCN amplification, the authors performed multivariate analysis and interestingly demonstrate an independent prognostic effect of PIM3 on neuroblastoma outcomes.

The authors demonstrate inhibition of Akt phosphorylation in an expanded panel of cell lines supporting the assertion that INL-302 (and related compounds) are capable of inhibiting these pathways. Unfortunately, due to technical difficulties, they were not able to show direct effects on PIM or downstream targets. They also more definitively demonstrate induction of apoptosis and differentiation in response to treatment with their kinase inhibitors. Additionally, these compounds appear to decrease N-Myc protein levels in the cells. Given the prominent role of N-Myc in neuroblastoma, this could be very beneficial. In an in vivo mouse model, IBL-302 is shown to inhibit tumor growth and prolong survival suggesting activity in neuroblastoma. More importantly, combining IBL-302 with commonly-used chemotherapy agents improved the effects of both drugs even leading to growth suppression. These conclusions are strengthened by the accompanying isobolographic analysis. The suggestion from the authors is that patients with neuroblastoma could obtain similar benefits while using less chemotherapy when combined with IBL-302. Given the high toxicity associated with neuroblastoma treatment, this would be a welcome advance.

Overall, the article is clear and well written. Once again, the authors outline their rationale for pursuing these studies is well laid out and easy to follow. I appreciate that the authors have taken the time to address the comments from the reviewers and augmented their data as requested. They have increased their references in regard to PI3K/mTOR pathway in neuroblastoma. The addition of more cell lines in the response assays helps demonstrate that they are not seeing cell-line specific effects. Unfortunately, the direct effect of IBL-302 on PIM and the downstream targets could not be measured directly, but not all proteins are amenable to Western analysis. I also liked the added experiment combining the targeted inhibitors (figure EV3), which showed some improved effect of the combination but not to the level seen with IBL-302. Also, by adding the Annexin V data, the

apoptosis effects are more convincing. I appreciated the addition of the isobolographic analysis, which supports their conclusions regarding combination chemotherapy. Finally, they have softened the language (especially in the discussion) so it now aligns nicely with the data presented. In total, I feel the changes to the revised version strengthened the paper.

We thank the reviewer for this nice summary of our paper and revised data and appreciate that the reviewer finds our amendments appropriate and interesting.

Minor Comment

1. In figure 6B, the addition of IBL-302 to low-dose cisplatin appears to have a synergistic effect in the LU-NB-3 and SK-N-BE(2)c cells. However, the SK-N-SH cells have a much different pattern of response and appear to possibly have an antagonistic effect. Do the authors have a possible cause for the different pattern seen in this cell line?

The response to IBL-302 and cisplatin combination treatment in SK-N-SH is indeed different than for the other cell lines. At the two lowest concentrations of cisplatin, there is a synergistic and additive effect, respectively. At the higher concentrations there might be an antagonistic effect, which, however, seems to be lost again at the highest concentration.

One explanation is that LU-NB-3 and SK-N-BE(2)c cells are MYCN-amplified cells and SK-N-SH is a non-MYCN-amplified cell line. It is possible that PIM/PI3K targeting has stronger effects against MYCN-amplified neuroblastoma as compared to non-MYCN-amplified tumors. We have emphasized this aspect in the manuscript.

Corresponding Author Name: Sofie Mohlin and Daniel Bexell

Manuscript Number: EMM-2018-10058-T